# SELF-ORGANIZED POLYNOMIAL-TIME COORDINATION GRAPHS

## ABSTRACT

Coordination graph is a promising approach to model agent collaboration in multi-agent reinforcement learning. It factorizes a large multi-agent system into a suite of overlapping groups that represent the underlying coordination dependencies. One critical challenge in this paradigm is the complexity of computing maximum-value actions for a graph-based value factorization. It refers to the decentralized constraint optimization problem (DCOP), which and whose constant-ratio approximation are NP-hard problems. To bypass this fundamental hardness, this paper proposes a novel method, named Self-Organized Polynomial-time Coordination Graphs (SOP-CG), which uses structured graph classes to guarantee the optimality of the induced DCOPs with sufficient function expressiveness. We extend the graph topology to be state-dependent, formulate the graph selection as an imaginary agent, and finally derive an end-to-end learning paradigm from the unified Bellman optimality equation. In experiments, we show that our approach learns interpretable graph topologies, induces effective coordination, and improves performance across a variety of cooperative multi-agent tasks.

## 1 INTRODUCTION

Cooperative multi-agent reinforcement learning (MARL) is a promising approach to a variety of real-world applications, such as sensor networks (Zhang & Lesser, 2011; Ye et al., 2015), traffic light control (Van der Pol & Oliehoek, 2016), and multi-robot formation (Alonso-Mora et al., 2017). One of the long-lasting challenges of cooperative MARL is how to organize coordination for a large multi-agent system. Coordination graph (Guestrin et al., 2001) is a classical approach to represent coordination relations in the reinforcement learning framework. It decomposes a multi-agent system into a suite of overlapping factors. Each factor is a hyper-edge covering a subset of agents that may involve coordinated behaviors. More specifically, the joint value function of the multi-agent system is factorized to the summation of local value functions based on factors or edges. Such a graph structure is a particular type of inductive bias that encodes task-specific coordination dependencies. It is known that conducting appropriate inductive bias can both reduce the model complexity and improve the generalizability (Battaglia et al., 2018).

One fundamental problem for coordination graphs is the trade-off between the representational capacity of value functions and the computational complexity of policy execution. To obtain value functions with high expressiveness, an advanced approach, deep coordination graphs (DCG) (Böhmer et al., 2020), considers a static complete graph connecting all pairs of agents. This graph structure has high representational capacity in terms of function expressiveness but raises a challenge for computation in the execution phase. The greedy action selection over a coordination graph can be formalized to a decentralized constraint optimization problem (DCOP), finding the maximum-value joint actions (Guestrin et al., 2001; Zhang & Lesser, 2013; Böhmer et al., 2020). Note that the DCOP and its any constant-ratio approximation are NP-hard problems, especially for the complete graph used in DCG (Dagum & Luby,

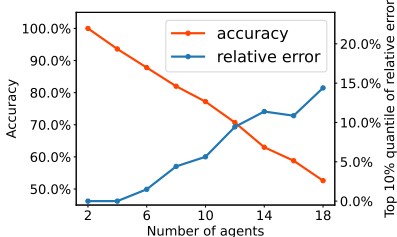

Figure 1: A motivating example with the accuracy and the relative joint Q error of max-sum algorithm w.r.t. the number of agents.

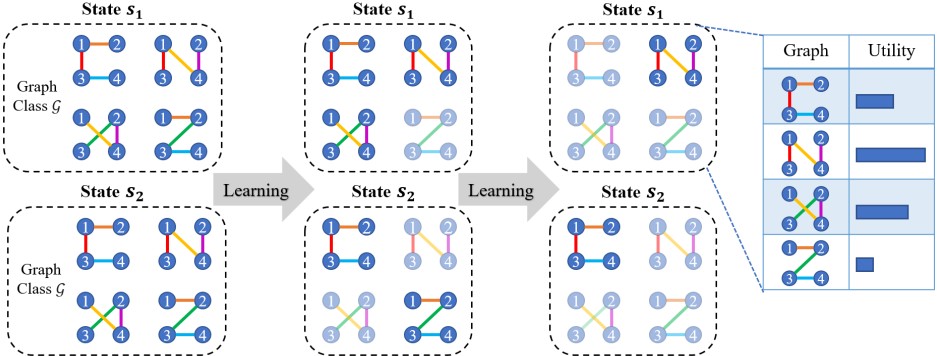

Figure 2: Illustration of our basic idea. For ease of viewing, we mark each edge with a unique color. Our algorithm conducts different coordination graphs for different environment states. For example, the selection of graph topologies on states $s_1$ and $s_2$ depends on the utility functions on corresponding states. These utility values are updated through the learning procedure.

1993; Park & Darwiche, 2004). From the perspective of theoretical computer science, any scalable heuristic algorithm (e.g., max-sum algorithm (Pearl, 1988)) for DCOP problem may have an unpredictable gap with the optimal solution. To visualize this phenomenon, we conduct a motivating example for generating a suite of complete-graph DCOPs with random edge-values. More detailed experiment setting are deferred to to Appendix A. We investigate the accuracy and the relative joint Q error of the max-sum algorithm in computing optimal joint actions, which is the default implementation of MARL methods based on coordination graphs (Stranders et al., 2009; Zhang & Lesser, 2013; Böhmer et al., 2020). The results are shown in Figure 1. As the number of agents increases, the accuracy of greedy action selection consistently decreases, and the relative joint Q error between the selected and optimal joint actions increases accordingly. How to incorporating polynomial-time complexity guarantee into coordination graphs with sufficient function expressiveness remains an open problem in MARL.

In this paper, we aim to overcome a dilemma in coordination graphs, i.e., expressive graph structures lead to computationally intractable DCOPs but concise graph structures are in lack of function expressiveness. To tackle this problem, we propose a novel graph-based MARL method for multi-agent coordination, named *Self-Organized Polynomial-time Coordination Graphs* (SOP-CG), that (1) utilizes structured graph topologies with polynomial-time guarantees for DCOPs and (2) extends their representation expressiveness through a dynamic graph organization mechanism. These two characteristics open up a new family of MARL algorithms. More specifically, we first construct graph classes guaranteeing polynomial-time DCOPs and then conduct a state-dependent graph selection mechanism. Such a dynamic factorization structure can be self-organized. We introduce an imaginary coordinator agent, whose action is to select a graph topology in a given class at each time step. SOP-CG incorporates this imaginary agent into the Bellman optimality equation supporting end-to-end learning. Figure 2 illustrates the basic idea of SOP-CG. We integrate the search of graph structures into the MARL paradigm. The utilities of different graph topologies are learned along with the RL objective. As the graph utilities are updated through agent-environment interactions, the coordination graph will evolve to the topology characterizing the coordination dependencies.

We evaluate the performance of SOP-CG in both grid-world and simulated physical environments. Empirical results on challenging tasks demonstrate that SOP-CG outperforms state-of-the-art baselines. By extensive ablation studies, we verify that harnessing specific structures with optimality guarantee of the induced DCOPs improves the sample quality and the accuracy of the one-step TD target. Furthermore, we show that SOP-CG can learn interpretable and context-dependent coordination graphs, which induces effective and dynamic coordination among agents.

## 2  RELATED WORK

Multi-agent reinforcement learning (OroojlooyJadid & Hajinezhad, 2019) is challenged by the size of joint action space, which grows exponentially with the number of agents. Independent Q-learning

(Tan, 1993; Foerster et al., 2017) models agents as independent learners, which makes the environment non-stationary in the perspective of each agent. An alternative paradigm called centralized training and decentralized execution (CTDE; Kraemer & Banerjee, 2016) is widely used in both policy-based and value-based methods. Policy-based multi-agent reinforcement learning methods use a centralized critic to compute gradient for the local actors (Lowe et al., 2017; Foerster et al., 2018; Wen et al., 2019; Wang et al., 2020d). Value-based methods usually decompose the joint value function into individual value functions under the IGM (individual-global-max) principle, which guarantees the consistency between local action selection and joint action optimization (Sunehag et al., 2018; Rashid et al., 2020b; Son et al., 2019; Wang et al., 2021a; Rashid et al., 2020a). Other work also studies this problem from the perspective of agent roles and individuality (Wang et al., 2020a;b; Jiang & Lu, 2021) or communication learning (Singh et al., 2018; Das et al., 2019; Wang et al., 2020c). Compared to these methods, our work is built upon graph-based value decomposition, which explicitly models the interaction among agents.

Coordination graphs are classical technique for planning in multi-agent systems (Guestrin et al., 2001; 2002b). They are combined with multi-agent deep reinforcement learning by recent work (Castellini et al., 2019; Böhmer et al., 2020; Li et al., 2020; Wang et al., 2021b). Joint action selection on coordination graphs can be modeled as a decentralized constraint optimization problem (DCOP), and previous methods compute approximate solutions by message passing among agents (Pearl, 1988). As the work which is most closed to our method, Zhang & Lesser (2013) presents an algorithm based on coordination graph searching. They define a measurement to quantify the potential loss of the lack of coordination between agents and search for a coordination structure to minimize the communication cost within restricted loss of utilities. However, their induced DCOPs still remain NP-hard. In contrast, minimizing communication is not our core motivation, and we aim to use a structured graph class to maintain sufficient function expressiveness when bypassing the computational hardness of large-scale DCOPs (Dagum & Luby, 1993; Park & Darwiche, 2004).

## 3 BACKGROUND

In this paper, we consider about fully cooperative multi-agent tasks that can be modelled as a Dec-POMDP (Oliehoek et al., 2016) defined as $\mathcal{M} = \langle D, S, \{A^i\}_{i=1}^n, T, \{O^i\}_{i=1}^n, \{\sigma^i\}_{i=1}^n, R, h, b_0, \gamma \rangle$, where $D = \{1, \ldots, n\}$ is the set of $n$ agents, $S$ is a set of states, $h$ is the horizon of the environment, $\gamma \in [0, 1)$ is the discount factor, and $b_0 \in \Delta(S)$ denotes the initial state distribution. At each stage $t$, each agent $i$ takes an action $a_i \in A^i$ and forms the joint action $\boldsymbol{a} = (a_1 \ldots, a_n)$, which leads to a next state $s'$ according to the transition function $T(s'|s, \boldsymbol{a})$ and an immediate reward $R(s, \boldsymbol{a})$ shared by all agents. Each agent $i$ observes the state only partially by drawing observations $o_i \in O^i$, according to $\sigma_i$. The joint history of agent $i$'s observations $o_{i,t}$ and actions $a_{i,t}$ is denoted as $\tau_{i,t} = (o_{i,0}, a_{i,0}, \ldots, o_{i,t-1}, a_{i,t-1}, o_{i,t}) \in (O^i \times A^i)^t \times O^i$.

**Deep Q-Learning.** Q-learning algorithms is a well-known algorithm to find the optimal joint action-value function $Q^*(s, \boldsymbol{a}) = r(s, \boldsymbol{a}) + \gamma \mathbb{E}_{s'}[\max_{\boldsymbol{a}'} Q^*(s', \boldsymbol{a}')]$. Deep Q-learning approximates the action-value function with a deep neural network with parameters $\boldsymbol{\theta}$. In Multi-agent Q-learning algorithms (Sunehag et al., 2018; Rashid et al., 2020b; Son et al., 2019; Wang et al., 2021a), a replay memory $D$ is used to store the transition tuple $(\boldsymbol{\tau}, \boldsymbol{a}, r, \boldsymbol{\tau}')$, where $r$ is the immediate reward when taking action $\boldsymbol{a}$ at joint action-observation history $\boldsymbol{\tau}$ with a transition to $\boldsymbol{\tau}'$. $Q(\boldsymbol{\tau}, \boldsymbol{a}; \boldsymbol{\theta})$ is used in place of $Q(s, \boldsymbol{a}; \boldsymbol{\theta})$, because of partial observability. Hence, parameters $\boldsymbol{\theta}$ are learnt by minimizing the following expect TD error:

$$\mathcal{L}(\boldsymbol{\theta}) = \mathbb{E}_{(\boldsymbol{\tau}, \boldsymbol{a}, r, \boldsymbol{\tau}') \in D} \left[ \left( r + \gamma V\left(\boldsymbol{\tau}'; \boldsymbol{\theta}^-\right) - Q\left(\boldsymbol{\tau}, \boldsymbol{a}; \boldsymbol{\theta}\right) \right)^2 \right] \tag{1}$$

where $V(\boldsymbol{\tau}'; \boldsymbol{\theta}^-) = \max_{\boldsymbol{a}'} Q(\boldsymbol{\tau}', \boldsymbol{a}'; \boldsymbol{\theta}^-)$ is the one-step expected future return of the TD target and $\boldsymbol{\theta}^-$ are the parameters of the target network, which will be periodically updated with $\boldsymbol{\theta}$.

**Coordination graphs.** An undirect coordination graph (CG, Guestrin et al. (2001)) $\mathcal{G} = \langle \mathcal{V}, \mathcal{E} \rangle$ contains vertex $v_i \in \mathcal{V}$ for each agent $1 \le i \le n$ and a set of (hyper-)edges in $\mathcal{E} \subseteq 2^{\mathcal{V}}$ which represents coordination dependencies among agents. Prior work considers higher order coordination where the edges depend on actions of several agents (Guestrin et al., 2002a; Kok & Vlassis, 2006; Guestrin et al., 2002b). Such a coordination graph induces a factorization of global Q function:

$$Q_{tot}(s, \boldsymbol{a}) = \sum_{v^i \in \mathcal{V}} f^i(a^i|s) + \sum_{\boldsymbol{e} = \{e_1, \ldots, e_{|e|}\} \in \mathcal{E}} f^{\boldsymbol{e}}(a^{e_1}, \ldots, a^{e_{|e|}}|s) \tag{2}$$

, where $f^i$ represents the individual utility of agent $i$ and $f^e$ specifies the payoff contribution for the actions of the agents connected by the (hyper-)edge $e$, so that the global optimal solution can be found through maximizing this joint value. The special case that $\mathcal{E}$ is an empty set yields VDN, but each additional edge enables the value representation of the joint actions of a pair of agents and can thus help to avoid relative-overgeneralization (Böhmer et al., 2020). In many coordination graph learning works (Zhang & Lesser, 2013; Böhmer et al., 2020; Wang et al., 2021b), the hyper-edges are simplified into pairwise edges. The graph is usually considered to be specified before training. Guestrin et al. (2002b) and Zhang & Lesser (2013) suggest that the graph could also depend on states, which means each state can have its own unique CG.

## 4 Self-Organized Polynomial-Time Coordination Graphs

A common method for multi-agent reinforcement learning is to decompose the joint value function into a linear combination of local value functions, each of which conditions on actions of a subset of agents (Guestrin et al., 2001; Böhmer et al., 2020). With this paradigm, computing joint action with maximum value can be modeled as a distributed constraint optimization problem (DCOP). General DCOP and any constant-ratio approximation have been proved to be NP-hard (Dagum & Luby, 1993; Park & Darwiche, 2004). Previous work adopts a variety of heuristic algorithms for action selection (Kok & Vlassis, 2006; Wang et al., 2021b). The imperfection of heuristics may lead to two side effects: (a) collecting bad samples and (b) errors in constructing one-step TD target, which hurt the learning performance.

To address these issues, we investigate *polynomial-time coordination graphs*, in which the induced DCOPs can be solved in polynomial time. We present a novel algorithm called *Self-Organized Polynomial-time Coordination Graphs* (SOP-CG), that utilizes a class of polynomial-time coordination graphs to construct a dynamic and state-dependent topology. In Section 4.1, we introduce our value factorization upon coordination graphs, which can hold the optimality guarantee of the induced DCOPs without significantly sacrificing the representational capacity by using a bottom-up design. To enable efficient topology self-organization, in Section 4.2, we unify the graph selection into the Q-learning framework by formulating the graphs as actions of an imaginary coordinator agent, and depict the whole framework of our algorithm. In Section 4.3, we further propose two polynomial-time graph class instances and discuss their theoretical advantages in SOP-CG.

### 4.1 Value Factorization on Polynomial-Time Coordination Graphs

To establish an algorithm with optimality guarantee in action selection, our approach first models coordination relations upon the graph-based value factorization specified by deep coordination graphs (DCG; Böhmer et al., 2020), in which the joint value function of the multi-agent system is factorized to the summation of *individual utility functions* $q_i$ and *pairwise payoff functions* $q_{ij}$ as follows:

$$Q(\boldsymbol{\tau}^{(t)}, \boldsymbol{a}; G) = \sum_{i \in [n]} q_i(\tau_i^{(t)}, a_i) + \sum_{(i,j) \in G} q_{ij}(\tau_i^{(t)}, \tau_j^{(t)}, a_i, a_j), \tag{3}$$

where the coordination graph $G$ is represented by a set of undirected edges. In our method, we focus on *Polynomial-Time Coordination Graph* which is defined as follows:

**Definition 1 (Polynomial-Time Coordination Graph)** *Let $\mathcal{G}_{Poly}$ denote the set of graph topologies whose induced DCOP can be solved in $Poly(n, \{|A_i|\}_{i=1}^n)$ running time with arbitrary edge values.*

In the literature of coordination graphs, we know that the set of undirected acyclic graphs $\mathcal{G}_{\text{uac}}$ is a subset of $\mathcal{G}_{\text{Poly}}$ (Fioretto et al., 2018). However, an undirected acyclic graph can contain at most $n-1$ edges in an environment with $n$ agents, which suffers from the lack of function expressiveness.

To alleviate this problem, our approach allows the graph topology to dynamically evolve through the transitions of environment status. Given different environmental states, the joint values can be factorized with different coordination graphs chosen from a predefined graph class $\mathcal{G} \subseteq \mathcal{G}_{\text{uac}} \subseteq \mathcal{G}_{\text{Poly}}$.

This design is based on an assumption that, although a long-horizon task cannot be characterized by a static sparse coordination graph, the coordination relations at each single-time step are sparse and manageable. The employment of this graph class fills the lack of representational capacity as well as maintains a polynomial complexity of the induced DCOPs.

## 4.2 Learning Self-Organized Topology with An Explicit Coordinator

Now we present a novel framework to render the self-organized coordination graph. We introduce an imaginary coordinator agent whose action space refers to the selection of graph topologies, aiming to select a proper graph for minimizing the loss within restricted coordination. When using the graph-based value factorization stated in Eq. (3), the graph topology $G$ can be regarded as an input of joint value function $Q(\boldsymbol{\tau}^{(t)}, \boldsymbol{a}; G)$. The objective of the coordinator agent is to maximize the joint value function over the specific graph class, which is a dual problem for finding a graph to minimize the loss within the different restricted coordination structures (Zhang & Lesser, 2013). Under this interpretation, we can integrate the selection of graph topologies into the trial-and-error loop of reinforcement learning. We handle the imaginary coordinator as a usual agent in multi-agent Q-learning framework and rewrite the joint action as $\boldsymbol{a}_{cg} = (a_1, \cdots, a_n, G)$.

**Execution.** Formally, at time step $t$, greedy action selection indicates the following joint action:

$$\boldsymbol{a}_{cg}^{(t)} \leftarrow \underset{(a_1, \cdots, a_n, G)}{\arg\max} \; Q(\boldsymbol{\tau}^{(t)}, a_1, \cdots, a_n; G). \tag{4}$$

Hence the action of coordinator agent $G^{(t)}$ is naturally the corresponding component in Eq. (4):

$$G^{(t)} \leftarrow \underset{G \in \mathcal{G}}{\arg\max} \left( \max_{\mathbf{a}} Q(\boldsymbol{\tau}^{(t)}, \boldsymbol{a}; G) \right). \tag{5}$$

After determining the graph topology $G^{(t)}$, the agents can choose their individual actions to jointly maximize the value function $Q(\boldsymbol{\tau}^{(t)}, \boldsymbol{a}; G^{(t)})$ upon the selected topology.

**Training.** With the imaginary coordinator, we can re-formulate the Bellman optimality equation and maximizing the future value over the coordinator agent's action:

$$Q^*(\boldsymbol{\tau}, \boldsymbol{a}; G) = \underset{\boldsymbol{\tau}'}{\mathbb{E}} \left[ r + \gamma \max_{G'} \max_{\boldsymbol{a}'} Q^*(\boldsymbol{\tau}', \boldsymbol{a}'; G') \right] \tag{6}$$

Since the graph selection is a part of agent action, the associative value $Q(\boldsymbol{\tau}, \boldsymbol{a}; G)$ of graph $G$ can be updated through temporal difference learning. The network parameters $\boldsymbol{\theta}$ can be updated by minimizing the standard Q-learning TD loss:

$$\mathcal{L}_{cg}(\boldsymbol{\theta}) = \underset{(\boldsymbol{\tau}, \boldsymbol{a}, G, r, \boldsymbol{\tau}') \sim \mathcal{D}}{\mathbb{E}} \left[ (y_{cg} - Q(\boldsymbol{\tau}, \boldsymbol{a}; G; \boldsymbol{\theta}))^2 \right] \tag{7}$$

where $y_{cg} = \max_{(\boldsymbol{a}', G')} Q(\boldsymbol{\tau}', \boldsymbol{a}'; G'; \boldsymbol{\theta}^-)$ is the one-step TD target and $\boldsymbol{\theta}^-$ are the parameters of target network. $G^{(t)}$ is regarded as part of the transition and is stored in the replay buffer together with the realistic agent actions. We include more implementation details of temporal difference learning in Appendix C.

The overall algorithm of our approach is summarized in Algorithm 1. The graph selection step at line 6 is the main characteristic of our method, which differs from the static graph representation used by prior work (Guestrin et al., 2002a; Kok & Vlassis, 2006; Böhmer et al., 2020). To support the self-organization of coordination graphs, the graph selection step raises a new computation demand, i.e., we need to find the best candidate from the graph class $\mathcal{G}$, which will be another non-trivial combinatorial optimization problem. We will show that, by designing a proper graph class $\mathcal{G}$, we can maintain both computational efficiency and function representational capacity in the next subsection.

## 4.3 Harnessing Structured Graph Classes for Efficient Coordination

To achieve the efficient graph search over the polynomial-time class instances, we investigate the following structures for the graph class $\mathcal{G}$, which are subsets of $\mathcal{G}_{\text{uac}}$:

---

**Algorithm 1** Self-Organized Polynomial-Time Coordination Graphs

---

**Require:** Predefined graph class $\mathcal{G}$
 1: Initialize replay buffer $\mathcal{D}$
 2: Initialize network with random weights
 3: **for** episode$= 1, \cdots, M$ **do**
 4:     Receive initial observation $\boldsymbol{\tau}^{(0)}$
 5:     **for** $t = 0, \cdots, T$ **do**
 6:         Choose coordination graph $G^{(t)} \in \mathcal{G}$ according to Eq. (5)        $\triangleright$ Section 4.3
 7:         Select actions $\boldsymbol{a}^{(t)}$ w.r.t $G^{(t)}$ and an $\epsilon$-greedy exploration
 8:         Execute actions $\boldsymbol{a}^{(t)}$ and receive reward $r^{(t)}$ and observation $\boldsymbol{o}^{(t+1)}$
 9:         Store transition $(\boldsymbol{\tau}^{(t)}, \boldsymbol{a}^{(t)}, r^{(t)}, \boldsymbol{\tau}^{(t+1)})$ in $\mathcal{D}$
10:     **end for**
11:     Sample random minibatch of transitions from $\mathcal{D}$
12:     Perform a gradient descent step on loss defined in Eq. (7)
13: **end for**

---

- *Pairwise grouping $\mathcal{G}_P$.* This class is introduced by Castellini et al. (2019). $n$ agents are partitioned to $\lfloor n/2 \rfloor$ non-overlapping pairwise groups. Only pairwise interactions are considered and two agents are connected with an edge if they lie in the same group.

- *Hierarchical organization $\mathcal{G}_H$.* Overlapping edges are allowed in this class. In a system with $n$ agents, the graph is organized as a tree with $n - 1$ edges so that all agents form a connected component. Even if two agents are not directly connected by an edge, their actions are implicitly coordinated through the path on the tree. This provides the potential to approximate complex joint value functions.

$\mathcal{G}_H$ is actually the set of maximal undirected acyclic graphs. It is noteworthy that we do not further incorporate the smaller graphs into both classes, since their representational capacity are dominated by existing graphs in the classes. We summarize the hardness of computing these problems upon different graph class in Table 1. Although the representational capacity of $\mathcal{G}_P$ is relatively weaker, selecting $G^{(t)}$ from this class can be rigorously solved by polynomial-time algorithms (Edmonds, 1965), which highlights the key benefit of $\mathcal{G}_P$. In comparison, selecting $G^{(t)}$ from $\mathcal{G}_H$ is non-trivial, so we design a greedy algorithm to compute approximated

| Graph Class | Select graph $G^{(t)}$ | Select actions on a given graph |
|:---:|:---:|:---:|
| $\mathcal{G}_P$ | $\checkmark$ | $\checkmark$ |
| $\mathcal{G}_H$ | $-$ | $\checkmark$ |
| Complete graph | N/A | $-$ |

Table 1: Computational considerations in choosing graph class. A check mark "$\checkmark$" means that known algorithms for the problem are guaranteed to find the optimal solution in polynomial time. The mark "$-$" denotes that these settings use the specific heuristic methods.

solutions. Note that, although the graph selection of $\mathcal{G}_H$ may be imprecise, its represented values are formally optimized by temporal difference learning with accurate action selection. Thus, as we will show in the experiment section, $\mathcal{G}_H$ can still work as a promising graph class in absence of rigorous graph selection. Please refer to Appendix B for the implementation details of the induced combinatorial optimization for graph selection.

**Remark.** In this paper, SOP-CG trades off the representational capacity of value function with the computational complexity within the polynomial-time boundary. Compared to fully connected graph, our sparse graphs restrict the value function class to a smaller space. This polynomial-time graph class enables SOP-CG to achieve accurate greedy action selection using limited message passing along the edges and achieve effective learning in the centralized training process. Moreover, if these graph classes are enough to express the coordination dependencies in a multi-agent task, using such topologies will be more sample-efficient by virtue of the concise function classes (Battaglia et al., 2018), which becomes another potential advantage of our algorithm.

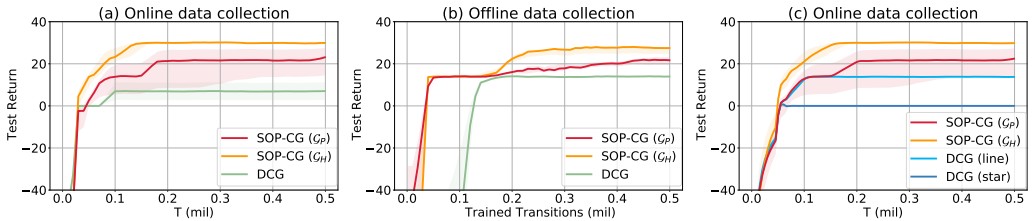

Figure 3: Ablation studies on Sensor-network: (a) comparison against DCG with online data collection; (b) comparison against DCG with offline dataset; (c) comparison with static topologies.

# 5    EXPERIMENTS

In this section, we conduct experiments to answer the following questions: (1) Does the class of polynomial-time coordination graphs improve our performance (see Section 5.1)? (2) How well does SOP-CG perform on complex cooperative multi-agent tasks (see Fig. 5)? (3) Can SOP-CG extract interpretable dynamic coordination structures (see Fig. 6)?

## 5.1    DIDACTIC EXAMPLE

We first study a didactic example to demonstrate two main contributions of this paper: (a) harnessing polynomial-time coordination graphs to obtain optimality guarantee of the induced DCOPs; (b) maintaining function expressiveness by incorporating a class of structured graphs into the Q-learning framework. We conduct ablation studies on *Sensor-network*, which is a classic task for testing multi-agent coordination (Lesser et al., 2003; Zhang & Lesser, 2011). We adopt the same task configuration as Wang et al. (2021b). As illustrated in Fig. 4, 15 sensor agents are arranged as a $3 \times 5$ ma-

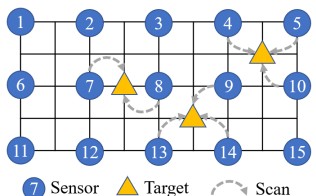

Figure 4: Sensor-network.

trix while three targets randomly walk in the same grid space. At each timestep, an agent can choose to scan one of the eight nearby positions (inducing a cost) or perform no operation. If two or more agents scan the target simultaneously, the target is captured and produces a positive joint reward. We compare our algorithm with DCG (Böhmer et al., 2020) on this game. The default implementation of DCG uses a fully connected coordination graph. In addition, DCG provides alternative implementations on two static graphs *Line* (agent $i$ is connected with agent $i-1$ and $i+1$) and *Star* (agent 1 is linked with others), which are two special instances of $\mathcal{G}_H$. SOP-CG ($\mathcal{G}_H$) has absolutely higher function expressiveness than DCG (line) and DCG (star).

For the first contribution, we verify that the optimality guarantee of the induced DCOP in our algorithm improve the performance. We test these algorithms in both online (Fig. 3a) and offline learning (Fig. 3b) settings. For fair evaluation, we train DCG (line) and collect 15K experienced episodes throughout the training process to set up the diverse offline dataset. As shown in Fig. 3(a-b), SOP-CG outperforms DCG in both settings. In the offline setting, as the dataset is fixed for both algorithms, the main difference between SOP-CG and DCG lies in the construction of one-step TD target. Our algorithm is guaranteed to find the optimal solution of the induced DCOPs, which must play an important role in reducing the errors in computing TD target. We also observe that DCG sticks in a bad local optimum with online data collection, but its performance is improved in the offline setting. This result indicates that the hardness of greedy action selection causes DCG to collect low-quality samples in the online setting, which may hurt the final performance. In contrast, our method converges to the same superior value in both settings, highlighting the effectiveness of polynomial-time coordination graphs.

For the second contribution, we show the importance of learning a state-dependent structure. As illustrated in Fig. 3c, the performance of SOP-CG is much better than both DCG (line) and DCG (star) with online data collection. A static sparse topology usually cannot express the coordination dependencies in all steps, so these methods could not work well even if they are built upon polynomial-time coordination graphs.

Figure 5: Learning curves on Pursuit, Chasing and Postman. The optimal return is demonstrated if it is available.

## 5.2 PERFORMANCE COMPARISON

To further demonstrate the effectiveness of our algorithm and discuss the difference among the graph classes, we evaluate SOP-CG on three large-scale complex tasks with highly dynamic coordination requirements and partial observation: Pursuit, Chase and Postman.

- *Pursuit*. This environment is also called predator-and-prey, which is widely used in recent work (Son et al., 2019; Böhmer et al., 2020; Wang et al., 2021b). 20 agents and 10 random walking prey interact on a $10 \times 10$ grid-world. Based on a limited sight range, an agent can move on the map or catch adjacent prey. One prey is captured only if at least two adjacent agents catch it simultaneously, inducing a joint reward of 1. After that, both agents and the captured prey are removed from the game. If only one agent tries to catch the prey, it will fail and receive a punishment of -1.

- *Chase*. This is a challenging task on the particle world with trained adversaries instead of built-in AIs created by Lowe et al. (2017). On the map with 3 randomly generated obstacles, 10 slower agents chase 3 faster adversaries. For each collision between an agent and an adversary, the agents receive a reward while the adversaries are punished. An agent or adversary can observe the units in a restricted circular area that is centered at its position. We train the agents by a specific algorithm, and the adversaries are concurrently trained by VDN.

- *Postman*. 12 postmen work in a vast community to deliver letters from residents to the post office. With the post office located in the northwest of the community, the whole community is separated into three parts according to the distance to the post office, and each postman will only work in one region. Postmen need to pick randomly generated letters in the community. While two postmen meet each other, any one of them can offer its letters to another one, so they can collaborate in order to deliver the letters to the post office. A postman can observe the information of other postmen and the letters in a sight range.

We compare our method with the state-of-the-art coordination graph learning methods (DCG (Böhmer et al., 2020), CASEC (Wang et al., 2021b)), fully decomposed value-based methods (VDN (Sunehag et al., 2018), QMIX (Rashid et al., 2020b)) and communication-based method (NDQ (Wang et al., 2020c)). The implementation details and hyperparameter settings are included in Appendix D. As presented in Fig. 5, SOP-CG outperforms the baselines in all three environments.

Using a smaller graph class will be more efficient if such a concise graph class is sufficient to express the coordination dependency in the task of interest. Taking the task Chase as an example, DCG and CASEC performs as poorly as the fully-decomposed methods, as their graph classes are too large and complex to learn. However, our method significantly outperforms coordination graph learning algorithms with a smaller graph class, which strongly confirms our statement. Furthermore, SOP-CG ($\mathcal{G}_H$) has better performance than SOP-CG ($\mathcal{G}_P$) on this task, which may be due to the lack of representational capacity of $\mathcal{G}_P$. On the contrary, Pursuit mainly involves pairwise cooperation, so SOP-CG ($\mathcal{G}_P$) achieves better sample efficiency than SOP-CG ($\mathcal{G}_H$).

We will show that reducing the error of DCOP can also improve performance. On task Pursuit with 20 agents, our method achieves almost the optimal of this task while DCG has over two times larger gap with the optimal value than ours. The reason is that Pursuit requires precise collaborations. If only one agent tries to catch the prey, it will fail and induce a large punishment. When facing growing scales, DCG and CASEC have a poor accuracy on DCOP, as we showed before, which may lead

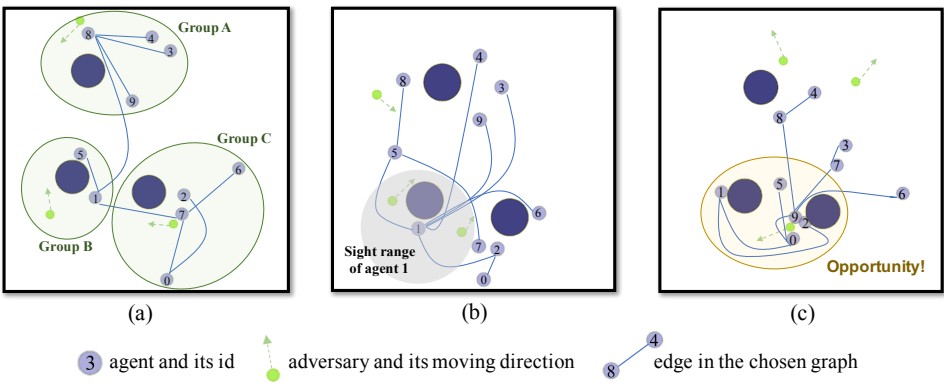

Figure 6: Various coordination graphs learned by our algorithm on Chase: (a) Self-organized grouping at initialization; (b) Connecting to agent with rich observation for better information sharing; (c) Concentrated collaboration structure around an enclosed adversary.

to inaccurate greedy action selection, impeding DCG to better performance. In the super hard task Postman, which requires agents to collaborate during long horizon, our method also outperforms other methods.

### 5.3 VISUALIZATION OF SELF-ORGANIZED COORDINATION

A major strength of our method is that SOP-CG can organize different coordination graphs to adapt to different situations. Such an adaptive organization mechanism can improve the efficiency of multi-agent collaboration. To illustrate the coordination relations organized by SOP-CG, we visualize the learned graph structures on Chase. In Fig. 6, we take three snapshots chronologically from one episode. This episode is collected by SOP-CG ($\mathcal{G}_H$) using the tree-based graph class. It shows that SOP-CG can produce interpretable graph structures that characterize the ground-truth coordination dependencies. The learned coordination behaviors are interpreted as follows:

1. Fig. 6a presents the initial status of the game, where agents and adversaries are generated randomly on the map. As the three adversaries distribute in separate positions, the agents naturally form three groups to chase different adversaries.

2. After several time steps, in Fig. 6b, two adversaries gather at the bottom left of the map. Note that, in this game, each agent can only observe objects within a small circular range. Only agent 1 can observe the two adversaries simultaneously, which makes it connect to the majority of the rest of the agents and helps them recognize the position of the adversaries.

3. Finally, as illustrated by Fig. 6c, the bottom adversary is surrounded by multiple agents, yielding an opportunity for the agents to win considerable rewards, which makes the collaboration structure concentrate around the adversary.

As presented above, the graph structures learned by SOP-CG can learn effective coordination patterns in Chase, which can be explained by the oracle collaboration strategy of human experts. It demonstrates the ability of our approach to organizing coordination relations.

## 6 CONCLUSION

This paper introduces SOP-CG, a novel multi-agent graph-based method that guarantees the polynomial-time greedy policy execution with expressive function representation of a structured graph class. To enable end-to-end learning, SOP-CG introduces an imaginary agent to dynamically select the state-dependent graph and incorporates it into a unified Bellman optimality equation. We demonstrate that SOP-CG can learn interpretable graph topologies and outperform state-of-the-art baselines on several challenging tasks. Although the graph class is exponentially large, we conduct two specific polynomial-time graph class instances to achieve impressive performance. Designing effective graph exploration methods to deal with larger graph classes will be an interesting future direction.

ETHICS AND REPRODUCIBILITY STATEMENTS

The main contribution of our work locates on the research of a foundation theoretical problem in optimality theory and try to improve the problem with our method. Our work does not have a bad social impact. The source code of our method for all the experiments is attached in supplementary material, and the settings and parameters for all models and algorithms mentioned in the experiment section please refer to Appendix D.

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

## A  EXPERIMENTAL DETAILS OF THE MOTIVATING EXAMPLE

**Algorithms.**  To get the correct result of DCOP, we use a brute force with complexity $O(|A|^n)$, while we choose Max-sum algorithm (has an alternative implementation called Max-plus) (Pearl, 1988), which is commonly used by most of the coordination graph learning methods (Zhang & Lesser, 2013; Böhmer et al., 2020; Wang et al., 2021b) to test the accuracy and relative Q error. The details of the algorithm are presented below (Zhang & Lesser, 2013).

Max-sum constructs a bipartite graph $\mathcal{G}_m = \langle \mathcal{V}_a, \mathcal{V}_q, \mathcal{E}_m \rangle$ according the coordination graph $\langle \mathcal{V}, \mathcal{E} \rangle$. Each node $v \in \mathcal{V}_m$ represents an agent who needs to do action selection, and each node $u \in \mathcal{V}_q$ represents a (hyper-)edge function. Edges in $\mathcal{E}_m$ connect $u$ with the corresponding agent nodes. Max-sum algorithm will do multi-round message passing on this graph. The number of iterations is usually set smaller than 10 in previous work, while we set it 100 in this experiment.

Message passing on this bipartite graph starts with sending messages from node $v \in \mathcal{V}_a$ to node $u \in \mathcal{V}_q$:

$$m_{v \rightarrow u}(a_i) = \sum_{h \in \mathcal{F}_v \setminus u} m_{h \rightarrow v}(a_i) + c_{vu} \tag{8}$$

where $\mathcal{F}_v$ represents the node connected to $v$, and $c_{vu}$ is the normalizing factor preventing the value of messages from growing arbitrarily large. The message from node $u$ to $v$ is:

$$m_{u \rightarrow v}(a_i) = \max_{\boldsymbol{a}_u \setminus a_v} \left[ f(\boldsymbol{a}_u) + \sum_{h \in \mathcal{V}_u \setminus v} m_{h \rightarrow u}(a_h) \right] \tag{9}$$

where $\mathcal{V}_u$ is the set of nodes connected to node $u$, $\boldsymbol{a}_u = \{a_h | h \in \mathcal{V}_u\}$, $\boldsymbol{a}_u \setminus a_v = \{a_h | h \in \mathcal{V}_u \setminus \{v\}\}$, and $f$ represents the value function of the (hyper-)edge. After iterations of message passing, each agent $v$ can find its optimal action by calculating $a_v^* = \operatorname{argmax}_{a_v} \sum_{h \in \mathcal{F}_v} m_{h \rightarrow v}(a_v)$.

**Test data.**  We take $1,000$ random seeds to test the problem. We fix the number of actions 3 and move the number of agents from 2 to 18. For each setting, we generate a fully connected graph with each value a uniformly random number in $\{-1, 0, 1\}$ along with a noise drawn from the normal distribution.

## B  COMPUTATIONS WITH CLASSES $\mathcal{G}_P$ AND $\mathcal{G}_H$

Here we present the details of algorithms we used to select graph $G^{(t)}$ from the graph class $\mathcal{G}$.

**When $\mathcal{G} = \mathcal{G}_P$, pairwise grouping.**  In this case, $n$ agents are pairwise matched by $\lfloor n/2 \rfloor$ edges. One agent will be isolated if $n$ is an odd number. With such a coordination graph, each agent coordinates with exactly one agent, except for the isolated one.

If node $i$ is matched with node $j$, it will contribute $q_{i \rightarrow j} = q_i(\tau_i, a_i) + q_j(\tau_j, a_j) + q_{ij}(\tau_i, \tau_j, a_i, a_j)$ to the total value when choosing action $a_i$, which only contains the single utility function and one pairwise function. Thus the pair $(i, j)$ can contribute at most

$$\max_{a_i, a_j} \left[ q_i(\tau_i, a_i) + q_j(\tau_j, a_j) + q_{ij}(\tau_i, \tau_j, a_i, a_j) \right] \tag{10}$$

to the joint value without effect the action selection of other agents.

Hence, we construct a undirected weighted graph $G = \langle \mathcal{V}, \mathcal{E}_w \rangle$, where $\mathcal{V}$ is the vertex set, and

$$\mathcal{E}_w = \left\{ \left( i, j, w(i, j) = \max_{a_i, a_j} \left[ q_i(\tau_i, a_i) + q_j(\tau_j, a_j) + q_{ij}(\tau_i, \tau_j, a_i, a_j) \right] \right) \middle| \forall (i, j) \in \mathcal{E} \right\}. \tag{11}$$

A matching on such graph is a function $mate(v)$ where $mate(mate(v)) = v$ for all $v \in \mathcal{V}$ with at most one $v$ satisfies that $mate(v) = v$. The weight of the matching is defined as $\sum_v w(v, mate(v))$. Our goal is to find the maximum weighted matching on this graph.

This problem can be solved by the blossom algorithm (Edmonds, 1965) in $O(n^3)$ time. The brief steps of this algorithm are to iteratively find augmentation paths on the graph and contract the odd-cycle, which is called blossom in this algorithm.

**When $\mathcal{G} = \mathcal{G}_H$, hierarchical organization**. In this case, the graph is an acyclic graph with $n-1$ edges, and all agents form a connected component. The distributed constraint optimization problem (DCOP) on such a graph can always be solved in the polynomial time (Fioretto et al., 2018).

Choosing the best graph $G^{(t)}$ in $\mathcal{G}_H$ is non-trivial. Therefore, we present a greedy algorithm to select the graph from the graph class. The high level idea is to add edges greedily until all agents are connected. At each iteration, we add an edge across two different connected components to the coordination graph, which optimizes the increment of the maximal joint value. For an edge set $E \subseteq \mathcal{E}$, we define the value of a set $f(E)$ to be the result of DCOP on graph $\langle \mathcal{V}, E \rangle$. Initially we have $E$ is an empty set. We do the iteration $n-1$ times. In each iteration, we find

$$e = \underset{e \in \mathcal{E}, E \cup \{e\} \text{ is acyclic}}{\arg\max} f(E \cup \{e\}), \tag{12}$$

and add this edge to the current graph, i.e., $E \leftarrow E \cup \{e\}$. Once a graph from $\mathcal{G}_H$ is selected, the joint action selection can be computed via dynamic programming.

**Time complexity**. We summarize the time complexity (in worst case) of each component in Table 2. For SOP-CG ($\mathcal{G}_H$), we use pruning techniques in our implementation, so the real-time cost of selecting graph $G^{(t)}$ is usually much smaller than the complexity reported in Table 2. The detailed training time cost of SOP-CG and DCG can be found in Appendix D.

| Algorithm | Select graph $G^{(t)}$ | Select actions on a given graph |
|---|---|---|
| SOP-CG ($\mathcal{G}_P$) | $O(n^3)$ | $O(nA^2)$ |
| SOP-CG ($\mathcal{G}_H$) | $O(n^3 A^2)$ (heuristic) | $O(nA^2)$ |
| DCG | N/A | $O(kn^2 A^2)$ (heuristic) |

Table 2: Time complexity of SOP-CG and DCG. $n$ is the number of agents and $A = |\bigcup_{i=1}^{n} A^i|$. For DCG, $k$ is the number of iterations in max-sum.

## C   GRAPH RELABELING TECHNIQUE

As studied by Hasselt (2010), vanilla Q-learning already has a tendency to overestimate, since the TD target takes a max operator over a set of estimated action-values. Notice that our framework introduces an additional max operator over graphs in class $\mathcal{G}$, which will probably induce extra upward bias. To attenuate this problem, we use a graph relabeling technique to control the effects of overestimation errors. In our implementation, we modify the vanilla TD loss defined in Eq. (7) to the following form:

$$\mathcal{L}_{cg}(\boldsymbol{\theta}) = \underset{(\boldsymbol{\tau}, \boldsymbol{a}, r, \boldsymbol{\tau}') \sim \mathcal{D}}{\mathbb{E}} \left[ \left( y_{cg} - \max_{G} Q(\boldsymbol{\tau}, \boldsymbol{a}; G; \boldsymbol{\theta}) \right)^2 \right] \tag{13}$$

where $y_{cg}$ is the one-step TD target defined in Section 4.2 and the current value no longer takes the graph which is stored in the replay buffer with this transition. As $G$ is the graph with largest estimated value, this loss achieves a soft constraint for the following target:

$$\forall G, Q(\boldsymbol{\tau}, \boldsymbol{a}; G; \boldsymbol{\theta}) \leq y_{cg} \tag{14}$$

which helps our method avoid overestimation.

We conduct ablation studies on the tasks Pursuit, Chase and Postman to demonstrate the necessity of this component. The results illustrated in Figure 7. SOP-CG ($\mathcal{G}_H$) w/o graph relabeling is more likely to stick in suboptimal strategies and may even collapse on Pursuit. As analyzed before, these phenomenons are due to the dramatic overestimation caused by the additional max operator over graphs. By contrast, the performance of SOP-CG ($\mathcal{G}_H$) is significantly improved, highlighting the effectiveness of graph relabeling.

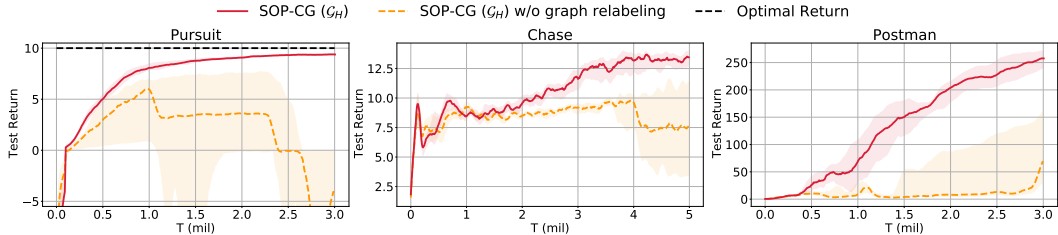

Figure 7: Ablation studies about the graph relabeling technique.

## D  IMPLEMENTATION DETAILS AND EXPERIMENT SETTINGS

A shared GRU (Cho et al., 2014) is used to process sequential inputs and then output the encoded observation history for each agent. A following fully-connected layer converts the observation histories to the individual utility function of each agent. The pairwise payoff function is computed by another multi-layer perceptron, which takes the concatenation of two agents' observation histories as input. Agents share parameters of the network for computing individual utility, and the parameters of the payoff network are also shared among different agent pairs.

All tasks in this paper use a discount factor $\gamma = 0.99$. We use $\epsilon$-greedy exploration, and $\epsilon$ anneals linearly from 1.0 to 0.05 over 50000 time-steps. We use an RMSProp optimizer with a learning rate of $5 \times 10^{-3}$ to train our network. A first-in-first-out (FIFO) replay buffer stores the experiences of at most 5000 episodes, and a batch of 32 episodes are sampled from the buffer during the training phase. The target network is periodically updated every 200 episodes.

Our method and all the baselines involved in this paper are implemented based on the open-sourced codebase PyMARL (Samvelyan et al., 2019), which ensures the fairness of comparison. For the baselines (VDN (Sunehag et al., 2018), QMIX (Rashid et al., 2020b), DCG (Böhmer et al., 2020), CASEC (Wang et al., 2021b)), we adopt the default hyperparameter settings provided by the authors. For evaluation, all learning curves in this paper are smoothed over 5 random seeds.

The experiments are finished on NVIDIA RTX 2080TI GPU. The training time of SOP-CG and DCG are listed in Table D. SOP-CG is much faster than DCG.

|  | Pursuit (3M timesteps) | Chase (5M timesteps) | Postman (3M timesteps) |
|---|---|---|---|
| Environmental time cost | 6h | 32h | 17h |
| SOP-CG ($\mathcal{G}_P$) | 19h | 46h | 31h |
| SOP-CG ($\mathcal{G}_H$) | 24h | 48h | 33h |
| DCG | 25h | 62h | 48h |

Table 3: Training time (hours) of SOP-CG and DCG. Environmental time cost denotes the total time taken by the environment simulator and the training of the adversaries.

## E  STARCRAFT II MICROMANAGEMENT BENCHMARK

In this section, we compare SOP-CG against DCG (Böhmer et al., 2020) and CASEC (Wang et al., 2021b) on StarCraft II micromanagement benchmark (Samvelyan et al., 2019). The pairwise payoff function has $A^2$ output head, where $A = |\bigcup_{i=1}^{n} A^i|$. DCG and CASEC adopt different techniques (e.g. low-rank approximation or learning action representations) to improve the learning efficiency of the payoff functions in environments with large action space. These techniques can also be applied to our algorithm. For fair evaluation, we test the vanilla versions of the algorithms without applying extra techniques on payoff networks. The results are shown in Fig. 8. SOP-CG outperforms DCG and CASEC on 4 of the 6 scenarios and achieves comparable performance on the other scenarios. This finding indicates that the class of polynomial-time coordination graphs is large enough to express coordinated policies in these tasks, and the accurate action selection further helps SOP-CG achieve higher peak performance than DCG and CASEC on the four maps.

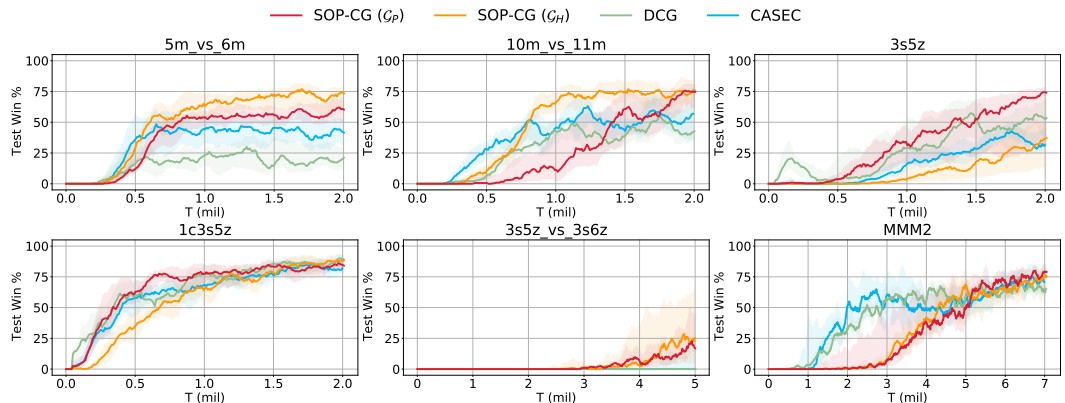

Figure 8: Learning curves on StarCraft II micromanagement tasks.

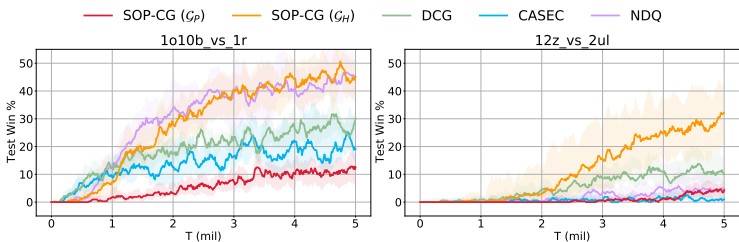

Figure 9: Learning curves on 1o10b_vs_1r and 12z_vs_2ul.

## F  SC2 TASKS RESTRICTED OBSERVABILITY

In coordination-graph-based methods, agents communicate to optimize joint action selection. As an alternative way, previous work studies multi-agent communication by learning message embedding (Singh et al., 2018; Das et al., 2019; Wang et al., 2020c). We compare our algorithm against the state-of-the-art communication learning method NDQ (Wang et al., 2020c) on 2 StarCraft II tasks: 1o10b_vs_1r and 12z_vs_2ul. These tasks are modified from the challenging maps introduced by the NDQ paper. The sight range of agents is reduced from 9 to 2, which enforces the agents to coordinate their actions by communication. To enable effective communication, we further augment the observation of each agent by its position on the map.

- 1o10b_vs_1r: 10 Banelings spawn randomly on a map full of cliffs and need to kill an enemy Roach. An allied Overseer detects the Roach and they spawn at the same random point. Therefore, the Banelings can get the position of the Roach from the allied Overseer.

- 12z_vs_2ul: 12 Zealots try to kill 2 powerful Ultralisk. All units spawn randomly on the map. The agents need to communicate the positions of enemies as well as coordinate the attack timing to reach high win rate.

The learning curves are shown in Figure 9. On the map 1o10b_vs_1r, a simple but effective communication strategy is to let the Overseer coordinate with all allied Banelings to guide their moving direction. We observe that it's not hard for NDQ to learn this mechanism and NDQ has a similar performance as SOP-CG ($\mathcal{G}_H$) on this map. However, the performance of NDQ is worse than the methods based on coordination graphs on 12z_vs_2ul. As all units spawn randomly on this map, the agents need to learn more complex coordinated strategies to kill both of the enemies. This finding suggests that directly coordinating actions based on coordination graphs may be more efficient than learning encoded messages. Notably, SOP-CG ($\mathcal{G}_H$) also outperforms DCG and CASEC on both tasks, highlighting the effectiveness of our method in complex scenarios.

## G  METRICS OF TASK COMPLEXITY

To illustrate the algorithmic properties of SOP-CG, we propose two measures of task complexity and conduct analysis between SOP-CG and DCG according to these measures. Furthermore, we design a new didactic example to empirically demonstrate our theoretical implications answering the following questions: when does SOP-CG work better and when does DCG work better?

We will introduce the two measures of task complexity as follows:

- **Problem size in general Dec-POMDPs.** Solving a general single-agent MDP with state space $S$ and action space $A$ requires the polynomial-time complexity of $|S| \times |A|$ (i.e., its problem size), as shown in Sutton et al. (Sutton & Barto, 2018). In MARL tasks, $A = \bigcup A_i$, whose problem size is extended to $|S| \times |A_i|^n$, exponential of the number of agents and the size of local action space, respectively. To analyze the effect of problem size on SOP-CG and CG, we find that the time complexity of exact greedy action selection (solving DCOPs) is exponential-time for DCG (i.e., with a complete graph), and which is polynomial-time complexity for SOP-CG (i.e., with a class of polynomial-time graphs).

- **Structural complexity of coordination within agents.** This measure is used to represent how much function expressiveness of coordination graph (i.e., which types of graph) is needed to solve a specific task. Note that the function capacity of DCG (line or star), SOP-CG, and DCG (complete graph) forms a hierarchy, i.e., $\mathcal{Q}_{\text{DCG (line or star)}} \subset \mathcal{Q}_{\text{SOP-CG}} \subset \mathcal{Q}_{\text{DCG (complete graph)}}$, where $\mathcal{Q}$ indicates the class of value function which can be represented by each algorithm. Thus, we formally design the structural complexity of coordination as the necessary and sufficient function capacity induced by each specific task. Along with this structural complexity increases, larger function capacity of algorithms is required. Moreover, we hypothesize that in the realistic tasks, the coordination structure is sparse (Kok & Vlassis, 2004; Zhang & Lesser, 2013) and the function class of SOP-CG would be sufficient for an extensive set of situations.

Using these two measures of task complexity, we compare SOP-CG with DCG in four cases that either these two complexities are high or low.

- **Both two complexities are low.** Both SOP-CG and DCG (complete graph) can easily address this case.

- **The problem size is large but the structural complexity is relatively low,** which is a common case in real-world tasks (Kok & Vlassis, 2004; Zhang & Lesser, 2013). SOP-CG can significantly outperform DCG (complete graph) because of efficient computation of exact greedy action selection without sacrificing its expressiveness in this case. The major strength of SOP-CG is its ability to model the value function using self-organized polynomial-time coordination graphs. Solving the DCOPs induced by SOP-CG is computationally efficient but which is expensive for DCG (complete graph) and the approximation of its induced DCOPs. We find that SOP-CG is not sensitive to the problem size, such as the number of agents and individual actions. In comparison, it is NP-hard to find a constant-ratio approximation of the DCOPs induced by complete graphs (i.e., the default implementation of DCG).

- **The problem size is small but the structural complexity is very high.** In this case, DCG (complete graph) may outperform SOP-CG as it is possible for DCG to compute nearly optimal joint action selection. However, as we hypothesized that this case may not usual in the realistic tasks since multi-agent systems are often large and the sparse coordination structure commonly exists in such systems.

- **Both two complexities are high.** SOP-CG and DCG (complete graph) may perform comparably in this case because they both introduce some approximation in learning. SOP-CG introduces approximation in value function expressiveness while DCG has inaccurate greedy action selection. This case is super hard and would be studied as an important future direction of coordination graph methods.

To investigate the metrics of task complexity and our performance relative to the metrics, we design a series of tasks. The tasks consist $m \times k$ agents separated into $m$ groups with each group $k$ agents.

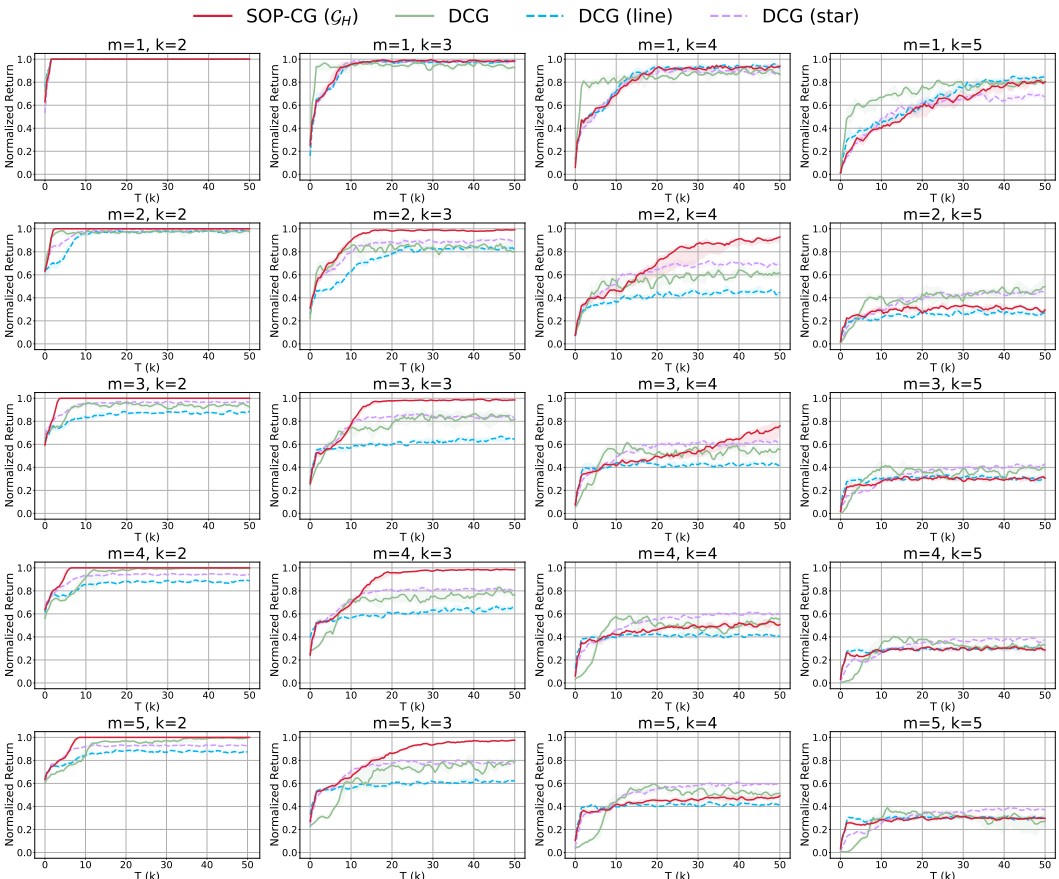

Figure 10: Learning curves on a series of toy games.

Each agent has to choose an action from the action space $\mathcal{A}$, and one group wins if and only if all its members perform $k$ distinct actions. The global reward is the proportion of winning groups.

In this task the number of agents in one group measures the representation complexity that each agent has to cooperate with other agents within the same group, while the number of groups indicates the scale of the problem. When the size of group grows, the coordination relationship becomes dense. If the number of groups increases and the size of group is fixed, the coordination complexity of the task will remain the same, but the scale of the task can be larger.

Results illustrated in Figure 10 show that our method can work very well when the group size of not so large (i.e., $k \leq 3$), being less sensitive to the number of groups. While DCG (line) or DCG (star) cannot work when the total number of agents $m \times k$ exceeds a small threshold, which corresponds to the fact that the representation capacity of them is restricted below our method. In contrast, the performance of our method decreases when the group size grows larger, while DCG performs better when $k$ grows and $m$ remains small. The result also suggests that when $k$ keeps small, both SOP-CG and DCG have enough representation capacity to solve the problem. As the scale of the problem grows, the complexity of solving DCOPs of DCG becomes higher, so SOP-CG can significantly outperform DCG.

## H SOME DETAIL DISCUSSIONS ABOUT TRADE-OFF IN SOP-CG

### H.1 CONNECTION TO A RECENT WORK ON HYPER-GRAPHS

We note that a recent work on the expressiveness of hyper-graphs (Yoon et al., 2020) is relevant to our work. The connections (i.e., the differences and relevance) between Yoon et al. (2020) and our paper are summarized as follows:

- **Different metric of expressiveness.** Yoon et al. (2020) investigates the functionality of graph expressiveness in terms of the order of hyper-edges. By contrast, our paper focuses on the expressiveness of different graph topologies.

- **Similar motivation.** Both Yoon et al. and our paper aim to study the trade-off between the graph expressiveness and the complexity of solving downstream tasks. In our paper, the downstream task specifically refer to the DCOPs induced by graph-based value factorization.

- **Different work flow and contributions.** Yoon et al. (2020) establish a systematic experiment study to provide an empirical guideline on how to trade off between the graph expressiveness and downstream costs. In comparison, our work proposes an algorithm to break this trade-off for improving MARL performance. Our algorithm is able to improve the function expressiveness without largely increasing the time complexity of downstream DCOPs.

Despite the expressiveness metric being different, several conclusions reached by Yoon et al. (2020) can be observed in our experiments. More specifically, the key findings of Yoon et al. (2020) (namely, three points, i.e, *Diminishing returns*, *Troubleshooter*, and *Irreducibility*) correspond to the following observations in our paper:

- **Diminishing returns.** When the task coordination structure is simple, a concise graph topology (e.g., pairwise matching) can achieve comparable performance with topologies using larger edge sets. As presented in Figure 5, SOP-CG ($\mathcal{G}_P$) performs better than SOP-CG ($\mathcal{G}_H$) and DCG in Pursuit, since pairwise matching is sufficient to express the underlying coordination relations in this task. (Recall that $\mathcal{G}_P$ uses pairwise matching and $\mathcal{G}_H$ uses tree-based topology.)

- **Troubleshooter and Irreducibility.** When the task requires a complicated coordination structure, the graph topology needs to have higher function expressiveness. This finding can also be seen in Figure 5. SOP-CG ($\mathcal{G}_H$) significantly outperforms SOP-CG ($\mathcal{G}_P$) in Chase where pairwise matching cannot express the underlying coordination relations.

### H.2 POSITION OF SOP-CG

The trade-off line from *DCG with fixed polynomial graphs (i.e., line or star)* to *SOP-CG* to *DCG with complete graph (i.e., the default configuration)* is similar to the *VDN-QMIX-QTRAN* methods, though SOP-CG lies in the coordination graph line of MARL (rather than the value factorization line). In this analogy hierarchy, SOP-CG corresponds to QMIX where SOP-CG has much larger function expressiveness than DCG with line or star graph (which corresponds to VDN), and can derive the exact greedy action selection compared to DCG (complete graph) (which corresponds to QTRAN). Note that different from VDN which has achieved excellent performance on StarCraft II benchmark (Samvelyan et al., 2019), DCG with line or star graph performs poorly due to its limited function capacity, which is mentioned in its original paper and in our evaluation (Appendix G). Similar to discussions in Mahajan et al. (2019) and Wang et al. (2021a) that QTRAN may suffer from its approximate greedy action selection, DCG may also suffer non-optimal action selection, while SOP-CG also considers the advantage of exact greedy selection (i.e., the property of polynomial-time graph proved by Fioretto et al. (2018)) and empirically achieves significant outperformance than DCG (Figure 5 and Appendix E and F). In comparison of function expressiveness, DCG (complete graph) achieves larger value function capacity than SOP-CG, but we hypothesize that in the realistic tasks, the coordination structure could be sparse (Kok & Vlassis, 2004; Zhang & Lesser, 2013) and the function class of SOP-CG will be sufficient to address a large suite of challenging tasks, e.g., StarCraft II, in Appendix E and F. Therefore, from the perspective of "VDN-QMIX-QTRAN" line in coordination graphs, SOP-CG has a clear and important position in the literature.

# I COMMUNICATION MECHANISM OF SOP-CG

SOP-CG runs in the paradigm of centralized training and decentralized execution with communication. We will discuss the communication mechanism in SOP-CG and its advantage comparing to the CTDE works as follows.

The communication mechanism of SOP-CG is consistent with DCG (Böhmer et al., 2020) and CASEC (Wang et al., 2021b). Two metrics are commonly used to measure the cost of multi-agent communication models: communication bandwidth and frequency. In SOP-CG, agents communicate the q-values to jointly decide the graph structure and induced actions, while DCG and CASEC use the message passing algorithms to optimize joint actions. This will require limited communication bandwidth as only a few real numbers are passed through each channel. We could further reduce the communication cost of SOP-CG by reducing the frequency of graph selection (i.e., changing the coordination graph via a fixed length of timesteps) if we assume the temporal consistency of coordination graph of the multi-agent system. This will be an interesting future direction for dynamic coordination graph in MARL.

Compared with the fully decentralized execution methods, the most important characteristic of coordination graph is the direct modeling of interactions between agent actions (i.e. pairwise payoff functions). The DCG paper (Böhmer et al., 2020) conducts comprehensive discussions to demonstrate the necessity of this design on tasks with relative overgeneralization pathology. Furthermore, it provides a straightforward objective for agent communication (i.e. maximizing the sum of individual utilities and pairwise payoffs), and the induced communication will promote agent cooperation especially in environments with partial observability (Wang et al., 2020c).

