# OpenReview forum: "Self-Organized Polynomial-time Coordination Graphs"
_ICLR.cc/2022/Conference — ICLR 2022 Submitted_

### Official Review · Reviewer_h3B4 · 2021-10-30

**Correctness:** 3
**Technical Novelty And Significance:** 3
**Empirical Novelty And Significance:** 3
**Recommendation:** 8
**Confidence:** 3

**Main Review:**

The detailed comments regarding the quality of this paper are listed as follows:
1. In the Introduction section, one of the biggest concerns is that this paper cannot tell the novelty of the proposed SOP-CG compared with the methods in peer researches. Besides, the illustration of experimental results is not convincible to prove the advantages of the proposed SOP-CG in this section.
2. Somewhere in the paper, the authors' presentation is unclear and is worth being improved. For example, in the Background section, the meanings of some symbols in the model are not clear. Furthermore, the authors should give the purposes of introducing formula (2).
3. What is the intuition behind the process of investigating polynomial-time coordination graphs in the Section 4? Or, is there any intuition at all? How did you come up with that idea? Have you borrowed this idea from somewhere else? It is better to give a detailed explanation about polynomial-time coordination graphs.
4. In page 6, the authors proposed Self-Organized Polynomial-Time Coordination Graphs. Whatever techniques are used in the manuscript, there is a need to tabulate computational cost of the proposed algorithm in this paper.
5. In the Experiments Sections, the authors claimed that the graph structures learned by SOP-CG definitely match the ground-truth demands for effective collaboration. However, it is not clear to us why the method can be used in demonstrating the ability of the proposed approach to organizing coordination relations. Therefore, the authors should provide a detailed explanation about the above issue.


**Summary Of The Paper:**

This paper introduced a novel method called Self-Organized Polynomial-time Coordination Graphs (SOP-CG), aiming to handle the decentralized constraint optimization problem (DCOP). This paper is well organized and the experiments are explicitly presented. Therefore, I think the work of this paper is very interesting and the contributions are sufficient.


**Summary Of The Review:**

The work presented is indeed interesting and relevant to the real scenarios. I recommend that this paper could be accepted.

---

> ### Author Response · Authors · 2021-11-23
> **Response to Reviewer h3B4 (Part II)**
>
>
> **Q3: What is the intuition behind the process of investigating polynomial-time coordination graphs in the Section 4? Or, is there any intuition at all? How did you come up with that idea? Have you borrowed this idea from somewhere else? It is better to give a detailed explanation about polynomial-time coordination graphs.**
>
> Our intuition is to trade off the representational capacity of value function with the computational complexity of greedy action selection within the polynomial-time boundary. The high-level idea of SOP-CG is original because we are very interested in the satisficing in AI and the exact greedy action selection of coordination graphs within the polynomial time complexity.
>
> We first observe that, in the literature of deep coordination graphs, SoTA baselines (e.g., DCG and CASEC) utilize heuristic algorithms (e.g., the max-sum algorithm [2]) to approximate the greedy action selections. Such a source of approximation error is systematic, since any constant-ratio approximation of DCOPs with the complete graph (which is the default configuration of DCG) is NP-hard [3]. This fact inspired us to study the polynomial-time coordination graph, whose definition is illustrated in Section 4 (see Definition 1):
>
> *Polynomial-time coordination graph is that its induced DCOPs can be solved in polynomial time.*
>
> However, fixed polynomial-time coordination graphs (e.g., line or star used in DCG) have restricted function expressiveness compared to the complete graph and empirically perform poorly, which is mentioned in DCG [4] and in our evaluation (see Appendix G). To enrich the value function capacity of polynomial-time graphs, we introduce the Self-Organized Polynomial-time Coordination Graphs (SOP-CG) and find that it can empirically achieve significant outperformance over DCG (Figure 5, Appendix E and F). This phenomenon strengthens our implicit hypothesis that in realistic tasks, coordination structure could be sparse [5, 6] and the function class of SOP-CG will be sufficient to address a large suite of challenging tasks, e.g., StarCraft II.
>
>
>
> **Q4: Whatever techniques are used in the manuscript, there is a need to tabulate computational cost of the proposed algorithm in this paper.**
>
> In the revised version, we discuss the worst-case time complexity of our algorithm in Appendix B. SOP-CG runs much faster than DCG, and the training time cost is listed in Table 3 (Appendix D). As pruning techniques are used in our implementations, the actual cost of SOP-CG is often much smaller than the worst-case time complexity.
>
> **Q5: The authors claimed that the graph structures learned by SOP-CG definitely match the ground-truth demands for effective collaboration.**
>
> Thanks for your careful comments. This is a typo and we update the statement and use the following claim in the revised paper: The graph structures learned by SOP-CG can learn effective coordination patterns in Chase, which can be explained by the oracle collaboration strategy of human experts.
>
> [1] Tonghan Wang, Jianhao Wang, Chongyi Zheng, and Chongjie Zhang.  Learning nearly decomposable value functions via communication minimization. In International Conference on Learning Representations, 2020.
>
> [2] Judea Pearl. Probabilistic Reasoning in Intelligent Systems: Networks of Plausible Inference. Morgan
> Kaufmann, 1988.
>
> [3] Paul Dagum and Michael Luby. Approximating probabilistic inference in bayesian belief networks
> is np-hard. Artificial intelligence, 60(1):141–153, 1993.
>
> [4] Böhmer, Wendelin, Vitaly Kurin, and Shimon Whiteson. "Deep coordination graphs." International Conference on Machine Learning. PMLR, 2020.
>
> [5] Kok, Jelle R., and Nikos Vlassis. "Sparse cooperative Q-learning." Proceedings of the twenty-first international conference on Machine learning. 2004.
>
> [6] Zhang, Chongjie, and Victor Lesser. "Coordinating multi-agent reinforcement learning with limited communication." Proceedings of the 2013 international conference on Autonomous agents and multi-agent systems. 2013.

---

> ### Author Response · Authors · 2021-11-23
> **Response to Reviewer h3B4 (Part I)**
>
> Thanks for the constructive comments. We provide clarification to your questions and concerns as below. If our response does not fully address your concerns, please post additional questions and we will be happy to have further discussions.
>
> **Q1.1: In the Introduction section, this paper cannot tell the novelty of the proposed SOP-CG compared with the methods in peer researches.**
>
> In this paper, we propose SOP-CG to tackle a critical trade-off in the literature of coordination graphs, i.e., the trade-off between function expressiveness and the complexity of induced DCOPs. In previous methods, using expressive graph structures leads to computationally intractable DCOPs but concise graph structures are in lack of function expressiveness. Our method, Self-Organized Polynomial-time Coordination Graphs (SOP-CG), is able to break this dilemma, which can improve the function expressiveness and keep the polynomial-time complexity within the greedy action and graph selections.
>
>
> To serve intuitions, we discuss an analogy to illustrate the position of SOP-CG in the literature of coordination graphs. The position of SOP-CG comparing to DCG's variants is similar to the development of VDN-QMIX-QTRAN methods, though SOP-CG lies in the coordination graph line of MARL (rather than the value factorization line). In this analogy hierarchy, SOP-CG corresponds to QMIX that interpolates between DCG using line/star and complete graphs. In comparison to DCG using line/star graphs (corresponding to VDN),  SOP-CG has much larger function expressiveness. In comparison to DCG using complete graph (corresponding to QTRAN), SOP-CG can derive the exact greedy action selection. In summary, SOP-CG enjoys both sides of advantages and addresses both sides of limitations. In experiments, we show that SOP-CG empirically achieves significant outperformance over DCG's variants (Figure 5, 8 and 10). From this perspective of "VDN-QMIX-QTRAN", SOP-CG has a clear significant position in the literature of coordination graph.
>
> The main contributions of this paper are summarized to two points: (1) introducing the critical role of polynomial-time coordination graphs in the theoretical and empirical perspectives, and (2) using self-organized state-dependent graph to enrich the function expressiveness of static polynomial-time graphs. Please refer to Section 5.1 for more empirical illustrations that justify our motivation and claims.
>
> In the rebuttal revision, we refine the discussion in section 1 to make the illustration of our contributions more clear.
>
>
>
> **Q1.2: The illustration of experimental results is not convincible to prove the advantages of the proposed SOP-CG in this section.**
>
> In the revised version, we compare SOP-CG against DCG and CASEC on StarCraft II micromanagement benchmark and the results are shown in Figure 8 (Appendix E). It is observed that SOP-CG demonstrates better learning efficiency and higher peak performance than DCG and CASEC on 4 of the 6 scenarios, and achieves comparable performance on the other scenarios. In Appendix F, we further conduct experiments on two SC2 maps with restricted observability. As shown in Figure 9, SOP-CG outperforms SoTA MARL baselines on both maps, in which the baselines are the MARL communication methods, NDQ [1], and coordination graph methods, DCG and CASEC. These findings indicate that the exact greedy action selection in our algorithm can both improve the efficiency of exploration and the accuracy of value computation, and self-organized state-dependent polynomial-time graph would be large enough to express coordinated policies in an extensive set of challenging tasks.
>
> **Q2: Somewhere in the paper, the authors' presentation is unclear and is worth being improved.**
>
> We are grateful for these valuable suggestions. The missing definitions have been added in the Background section. The purpose of introducing Eq. (2) is to show the most general definition of coordination graphs, which can decompose the joint utility based on a set of hyperedges. While in our work, we focus on a simplified version of coordination graphs with only pairwise edges.

---

### Official Review · Reviewer_xmMA · 2021-11-02

**Correctness:** 3
**Technical Novelty And Significance:** 2
**Empirical Novelty And Significance:** 2
**Recommendation:** 3
**Confidence:** 4

**Main Review:**

**Strengths**
1. To determine state-dependent coordination graph is interesting.
2. Incorporating graph selection into TD learning is desirable.

**Concerns/Questions**
1. State-dependent coordination graph needs to be determined at each time step in both training and execution,  which means SOP-CG is a centralized method. Thus, a baseline of single-agent RL is desired for comparison in experiments.
2. It is not clear how $q_i$ and $q_{ij}$ are learned. Are they parameter-sharing for agents?
3. The operation $\arg\max$ in equation 4 takes $O(n^3)$ for $\mathcal{G}_P $. It may be too costly for both training and execution. One evidence is that each run of SOP-CG  takes up to 2.5 days in these simple experimental tasks.
4. Since both DCG and CASEC include SMAC experiments, it would be better to also include it here to show the performance of SOP-CG in complex environments.

In summary, it is currently hard to see the benefit of determining the coordination graph in a centralized way. Moreover, the proposed method is not verified in complex environments.



**Minor Comments**
1. As claimed in Appendix C, a graph relabeling technique is used to solve the extra overestimation error introduced by additional max operator over graphs.  However, this paper is currently missing an ablation to validate this point.





**Summary Of The Paper:**

This paper proposes an extension of deep coordination graph, called Self-Organized Polynomial-time Coordination Graphs (SOP-CG). Instead of pre-specified graph topology used in DCG, their method allows graph topology to be state-dependent, which is achieved by a coordinator agent, and the optimization of this agent is incorporated in a modified temporal difference learning paradigm. Two pre-specified undirected acyclic graph classes are used to ensure polynomial-time graph selection and accurate greedy action selection. The result on sensor network, grid world and MPE shows that such a trade-off between the representational capacity of graph topology and the computational accuracy can improve the performance of MARL and learn meaningful graph topology.

**Summary Of The Review:**

State-dependent coordination graph is important. However, the paper currently has several weaknesses as mentioned above. It seems clearly below the bar of ICLR.

---

> ### Author Response · Authors · 2021-11-23
> **Response to Reviewer xmMA (Part II)**
>
> **Q3: The operation $\arg\max$ in equation 4 takes $O(n^3)$ for $\mathcal{G}_P$. It may be too costly for both training and execution.**
>
> We list the detailed time cost of training our algorithm in Table 3 (Appendix D) and find that SOP-CG runs much faster than DCG. The most time-consuming task is Chase with 10 agents, since the simulator needs to compute the physical collisions and force among 16 objects. SOP-CG spends 2 days and DCG spends 2.5 days to finish 5M steps training on this task, while the environmental time cost is about 1.5 days. We also discuss the worst-case time complexity of our algorithm in Appendix B. It is noteworthy that DCG takes $O(kn^2A^2)$ time for joint action selection ($k$ is the number of iterations in max-sum, $n$ and $A$ are the number of agents and actions, respectively), which is larger than $O(n^3)$ in usual cases.
>
> **Q4: Since both DCG and CASEC include SMAC experiments, it would be better to also include it here.**
>
> In the revised version, we compare SOP-CG against DCG and CASEC on StarCraft II micromanagement benchmark and the results are shown in Figure 8 (Appendix E). SOP-CG outperforms DCG and CASEC on 4 of the 6 scenarios and achieves comparable performance on the other scenarios. In Appendix F, we further conduct experiments on two SC2 maps with restricted observability. As shown in Figure 9, SOP-CG also achieves remarkable outperformance than the SoTA MARL baselines on both maps, in which the baselines are the MARL communication methods, NDQ [1], and coordination graph methods, DCG and CASEC.
>
> **Q5: This paper is currently missing an ablation about the graph relabeling technique.**
>
> We thank the reviewer for this suggestion. We add an ablation study in Appendix C and SOP-CG ($\mathcal{G}_H$) significantly outperforms the version without graph relabeling. As shown in Figure 7, SOP-CG ($\mathcal{G}_H$) w/o graph relabeling is more likely to stick in suboptimal strategies and may even collapse on Pursuit. These phenomenons are due to the dramatic overestimation caused by the additional max operator over graphs. The graph relabeling technique empirically addresses the overestimation problem and hence improves the performance of SOP-CG ($\mathcal{G}_H$).
>
> **Q6 (Summary): State-dependent coordination graph is important. However, the paper currently has several weaknesses as mentioned above.**
>
> We summarize our clarification to your major concerns as follows:
>
> - SOP-CG runs in the paradigm of centralized training and decentralized execution with communication, and we discuss the communication mechanism in Q1.1 and Appendix.
>
> - We conduct experiments on StarCraft II benchmark (Appendix E,F) in the revised version, and SOP-CG outperforms the baselines on 6 of the 8 maps. Please refer to Q4 for detailed discussions.
>
> - As discussed in Q3, SOP-CG runs much faster than DCG. We list the real-time training cost in Table 3 and further discuss the worst-case time complexity of our algorithm in Appendix B.
>
> If our response does not fully address your concerns, please post additional questions and we will be happy to have further discussions.
>
> [1] Tonghan Wang, Jianhao Wang, Chongyi Zheng, and Chongjie Zhang.  Learning nearly decomposable value functions via communication minimization. In International Conference on Learning Representations, 2020.
>
> [2] Böhmer, Wendelin, Vitaly Kurin, and Shimon Whiteson. "Deep coordination graphs." International Conference on Machine Learning. PMLR, 2020.

---

> ### Author Response · Authors · 2021-11-23
> **Response to Reviewer xmMA (Part I)**
>
> Thanks for the inspiring comments. We provide clarification to your questions and concerns as below. If our response does not fully address your concerns, please post additional questions and we will be happy to have further discussions.
>
> **Q1.1: SOP-CG is a centralized method.**
>
> Not exactly. SOP-CG lies in the paradigm of centralized training and decentralized execution with communication. We will clarify the communication mechanism in SOP-CG and its advantage comparing to the fully decentralized execution methods as follows.
>
> SOP-CG employs the same communication mechanism as DCG and CASEC. Two metrics are commonly used to measure the cost of multi-agent communication models: communication bandwidth and frequency. In SOP-CG, agents communicate the q-values to jointly decide the graph structure and induced actions, while DCG and CASEC use the message passing algorithms to optimize joint actions. This will require limited communication bandwidth as only a few real numbers are passed through each channel. We could further reduce the communication cost of SOP-CG by reducing the frequency of graph selection (i.e., changing the coordination graph via a fixed length of timesteps) if we assume the temporal consistency of coordination graph of the multi-agent system. This will be an interesting future direction for dynamic coordination graph in MARL.
>
> The most important characteristic of coordination graph is the direct modeling of interactions between agent actions (i.e. pairwise payoff functions), which is different from the fully decentralized execution methods. The DCG paper [2] conducts comprehensive discussions to demonstrate the necessity of this design on tasks with relative overgeneralization pathology. Furthermore, it provides a straightforward objective for agent communication (i.e. maximizing the sum of individual utilities and pairwise payoffs), and the induced communication will promote agent cooperation especially in environments with partial observability [1].
>
>
>
> **Q1.2: A baseline of single-agent RL is desired for comparison in experiments.**
>
> Single-agent RL algorithms are not applicable due to the large joint action space. For example, the task Pursuit involves 20 agents and each agent has 9 actions, yielding a joint action space of size $9^{20}\approx 1.2\times 10^{19}$. It is computationally intractable for DQN to handle such an exponential-size action space.
>
>
> In the revised version, we further compare our algorithm against the state-of-the-art MARL communication method NDQ [1]. This method allows agents to exchange information, which uses the same system assumption for message-passing as our algorithm. The results are updated in Figure 5 and Figure 9. SOP-CG significantly outperforms NDQ on all the 5 tasks.
>
>
> **Q2: It is not clear how $q_i$ and $q_{ij}$ are learned. Are they parameter-sharing for agents?**
>
> $q_i$ and $q_{ij}$ are learned end-to-end by the TD loss defined in Eq. (7). As described in Appendix D, agents share parameters of the network for computing individual utility, and the parameters of the payoff network are also shared among different agent pairs. This setting is consistent with DCG and CASEC.

---

### Official Review · Reviewer_Jx51 · 2021-11-08

**Correctness:** 3
**Technical Novelty And Significance:** 2
**Empirical Novelty And Significance:** Not applicable
**Recommendation:** 3
**Confidence:** 4

**Main Review:**

The main strength of the paper lies in its algorithm being of polynomial-time nature, and that characteristic needs much more emphasis in both text and figures. Does SOP-CG reach peak performance faster? Is its peak performance higher? could be some guideline questions when structuring the paper contents around the polynomial-time algorithm.

Q1. I would appreciate if the authors would discuss the relevance of a WWW 2020 paper: How Much and When Do We Need Higher-order Information in Hypergraphs? A Case Study on Hyperedge Prediction by Yoon et al. I think it could also provide some useful insights as to how high of an order the hyperedges need to be, to save some function expressiveness (i.e., representational capacity) at the cost of time complexity.

Q2. How did the authors go about designing the didactic examples? How simple of a task is the "sweet spot" for SOP-CG? As task complexity is altered, when does SOP-CG begin losing its simplicity advantage? When do other models take over? Which measures would you say most greatly determine the task complexity, which in turn, compromises SOP-CG's performance?

Q3. How would you position SOP-CG in the MARL research? Since the graph selector agent is there at both training at execution times, would SOP-CG be a centralized training centralized execution work? If it is, comparison against CTDE works such as VDN and QMIX would not be fair. It is as though a "free" centralized coordinator is helping out the SOP-CG agents at execution time as well. It may be a good idea to compare against some communication-enabled MARL works.

Q4. Despite starting out strong, the paper seems to fall off rather dramatically, especially when it comes to the didactic nature of the examples, tied in with the page 6 remark that SOP-CG would perform best (given its tradeoff) in tasks where the restricted graph classes are enough to express the coordination dependencies. This remark really sounds like going back on the VDN-QMIX-QTRAN line of research, whose focus was about covering a richer class of joint action-value functions. Going back on that trend now only to pursue the polynomial-time nature of the running algorithm would in my opinion require far more diverse evaluation examples, backed by a stronger motivation highlighting real-world threats of all the other MARL algorithms taking longer than polynomial time. As is, SOP-CG does not contend amazingly against other MARL algorithms that chose the "NP-hard? Curse of dimensionality? Fine. We'll approximate, approximate, approximate." path rather than the "Polynomial time is our topmost priority; function expressiveness can wait." path. That leads me back to the question of why pursue polynomial time at the cost of losing both the function expressiveness and the peak performance in the apparent trilemma.

**Summary Of The Paper:**

Authors present an imaginary coordinator agent that executes graph selection as its action based on possible combinations of pairwise edges and the underlying individual and pairwise utilities. The results show some effectiveness of the presented method, SOP-CG, in simple examples.

**Summary Of The Review:**

My biggest concern is the "imaginary" agent freely collecting information, making decisions, and delivering those decisions to all the agents at both the training and execution times. Comparison against the chosen baselines is not fair, even when the chosen evaluation task is a simple enough one, in which SOP-CG's limited representational capacity would not appear that pronounced.

---

> ### Author Response · Authors · 2021-11-23
> **Response to Reviewer Jx51 (Part V)**
>
> **Q5 (Summary): My biggest concern is the "imaginary" agent freely collecting information, making decisions, and delivering those decisions to all the agents at both the training and execution times. Comparison against the chosen baselines is not fair, even when the chosen evaluation task is a simple enough one, in which SOP-CG's limited representational capacity would not appear that pronounced.**
>
> We summarize our clarification to your major concerns as follows:
>
> - SOP-CG runs in the paradigm of centralized training and decentralized execution with communication, and we discuss the communication mechanism in Q3.1 and Appendix.
>
> - Our major baselines are previous methods based on coordination graphs, and we compare our method against the SoTA communication learning method (Figure 5, 9) in the revised paper. Both comparisons are sensible and fair. Experiments on StarCraft II benchmark (Appendix E,F) are added to support our claims. Please refer to Q3.2 for detailed discussions.
>
> - In Appendix G, we propose two measures of task complexity and discuss about relative advantages of SOP-CG and DCG based on these complexities. The claims are empirically verified by an elaborate didactic example. Please refer to Q2.2 for detailed discussions.
>
> If our response does not fully address your concerns, please post additional questions and we will be happy to have further discussions.
>
> References
>
> [1] Yoon, Se-eun, et al. ''How Much and When Do We Need Higher-order Information in Hypergraphs? A Case Study on Hyperedge Prediction.'' Proceedings of The Web Conference 2020. 2020.
>
> [2] Tonghan Wang, Jianhao Wang, Chongyi Zheng, and Chongjie Zhang.  Learning nearly decomposable value functions via communication minimization. In International Conference on Learning Representations, 2020.
>
> [3] Mikayel Samvelyan, Tabish Rashid, Christian Schroeder de Witt, Gregory Farquhar, Nantas Nardelli, Tim GJ Rudner, Chia-Man Hung, Philip HS Torr, Jakob Foerster, and Shimon Whiteson. The starcraft multi-agent challenge. In Proceedings of the 18th International Conference on Autonomous Agents and MultiAgent Systems, pp. 2186–2188. International Foundation for Autonomous Agents and Multiagent Systems, 2019.
>
> [4] Anuj Mahajan, Tabish Rashid, Mikayel Samvelyan, and Shimon Whiteson. Maven: Multi- agent variational exploration. In Advances in Neural Information Processing Systems, pages 7613–7624, 2019.
>
> [5] Jianhao Wang, Zhizhou Ren, Terry Liu, Yu Yang, and Chongjie Zhang. Qplex: Duplex dueling multi-agent q-learning. In International Conference on Learning Representations, 2021.
>
> [6] Sutton, Richard S., and Andrew G. Barto. Introduction to reinforcement learning. Vol. 135. Cambridge: MIT press, 1998.
>
> [7] Böhmer, Wendelin, Vitaly Kurin, and Shimon Whiteson. "Deep coordination graphs." International Conference on Machine Learning. PMLR, 2020.
>
> [8] Ferdinando Fioretto, Enrico Pontelli, and William Yeoh.  Distributed constraint optimization prob-lems and applications: A survey.Journal of Artificial Intelligence Research, 61:623–698, 2018.
>
> [9] Kok, Jelle R., and Nikos Vlassis. "Sparse cooperative Q-learning." Proceedings of the twenty-first international conference on Machine learning. 2004.
>
> [10] Zhang, Chongjie, and Victor Lesser. "Coordinating multi-agent reinforcement learning with limited communication." Proceedings of the 2013 international conference on Autonomous agents and multi-agent systems. 2013.

---

> ### Author Response · Authors · 2021-11-23
> **Response to Reviewer Jx51 (Part IV)**
>
> **Q4.1: This remark really sounds like going back on the VDN-QMIX-QTRAN line of research, whose focus was about covering a richer class of joint action-value functions.**
>
> Thanks for your inspiring comments. The trade-off line from *DCG with fixed polynomial graphs (i.e., line or star)* to *SOP-CG* to *DCG with complete graph (i.e., the default configuration)* is similar to the *VDN-QMIX-QTRAN* methods, though SOP-CG lies in the coordination graph line of MARL (rather than the value factorization line). In this analogy hierarchy, SOP-CG corresponds to QMIX where SOP-CG has much larger function expressiveness than DCG with line or star graph (which corresponds to VDN), and can derive the exact greedy action selection compared to DCG (complete graph) (which corresponds to QTRAN). Note that different from VDN which has achieved excellent performance on StarCraft II benchmark [3], DCG with line or star graph performs poorly due to its limited function capacity, which is mentioned in its original paper and in our evaluation (Appendix G). Similar to discussions in Mahajan et al. [4] and Wang et al. [5] that QTRAN may suffer from its approximate greedy action selection, DCG also suffers from non-optimal action selection, while SOP-CG has the advantage of efficient exact greedy selection (i.e., the property of polynomial-time graph proved by Fioretto et al. [8]) and empirically achieves significant outperformance than DCG (Figure 5 and Appendix E). In comparison of function expressiveness, DCG (complete graph) achieves larger value function capacity than SOP-CG, but we hypothesize that in the realistic tasks, the coordination structure could be sparse [9, 10] and the function class of SOP-CG will be sufficient to address a large suite of challenging tasks, e.g., StarCraft II, in Appendix E and F. Therefore, from the perspective of "VDN-QMIX-QTRAN" line in coordination graphs, SOP-CG has a clear significant position in the literature.
>
> **Q4.2: Why pursue polynomial time at the cost of losing both the function expressiveness and the peak performance in the apparent trilemma.**
>
> We respectfully disagree with this statement. SOP-CG does not lose the peak performance and its self-organized state-dependent polynomial-time graph would be large enough to express coordinated policies in an extensive set of challenging tasks. In the revised version, we conduct a large suite of experiments in StarCraft II to empirically illustrate the advantage of SOP-CG. We compare SOP-CG against DCG and CASEC (i.e., SoTA coordination graph baselines) on StarCraft II micromanagement benchmark in Figure 8 (Appendix E) and Figure 9 (Appendix F). SOP-CG significantly outperforms DCG and CASEC on 6 of the 8 scenarios and achieves comparable performance on the other scenarios. This observation indicates that the class of polynomial-time graphs can express coordinated policies in these tasks, and the exact greedy action selection on coordination graphs further helps SOP-CG achieve higher peak performance than DCG and CASEC on the 6 maps.

---

> ### Author Response · Authors · 2021-11-23
> **Response to Reviewer Jx51 (Part III)**
>
> **Q3.1: How would you position SOP-CG in the MARL research? A centralized training centralized execution work?**
>
> Not exactly. SOP-CG runs in the paradigm of centralized training and decentralized execution with communication. We discuss the communication mechanism in SOP-CG and its advantage compared to the CTDE works as follows.
>
> The communication mechanism of SOP-CG is consistent with DCG and CASEC. Two metrics are commonly used to measure the cost of multi-agent communication models: communication bandwidth and frequency. In SOP-CG, agents communicate the q-values to jointly decide the graph structure and induced actions, while DCG and CASEC use the message passing algorithms to optimize joint actions. This will require limited communication bandwidth as only a few real numbers are passed through each channel. We could further reduce the communication cost of SOP-CG by reducing the frequency of graph selection (i.e., changing the coordination graph via a fixed length of timesteps) if we assume the temporal consistency of coordination graph of the multi-agent system. This will be an interesting future direction for dynamic coordination graph in MARL.
>
> Compared with the fully decentralized execution methods, the most important characteristic of coordination graph is the direct modeling of interactions between agent actions (i.e. pairwise payoff functions). The DCG paper [7] conducts comprehensive discussions to demonstrate the necessity of this design on tasks with relative overgeneralization pathology. Furthermore, it provides a straightforward objective for agent communication (i.e. maximizing the sum of individual utilities and pairwise payoffs), and the induced communication will promote agent cooperation especially in environments with partial observability [2].
>
>
>
>
> **Q3.2: Comparison against CTDE works such as VDN and QMIX would not be fair. It may be a good idea to compare against some communication-enabled MARL works.**
>
> In the experiment section, our major baselines are previous methods based on coordination graphs (i.e., DCG and CASEC) as they run in the same nature as SOP-CG. This gives a fair comparison to demonstrate the strength of SOP-CG. The comparison to the start-of-the-art CTDE methods (i.e. VDN and QMIX) is for demonstrating advantages of CTDE+communication methods in partial observable environments and tasks with relative overgeneralization pathology.
>
> As suggested by the reviewer, we further compare our algorithm against the state-of-the-art communication learning method NDQ [2] in the revised paper. This method learns to pass encoded messages among agents, which uses the same system assumption as our algorithm. As shown in Figure 5, our method significantly outperforms NDQ on the three tasks. To facilitate further discussions, in Appendix F, we conduct experiments on two SC2 maps with restricted observability. It is observed that when the coordination requirement is simple (e.g. only one key agent needs to coordinate with others), NDQ learns effective communication and performs similarly to SOP-CG, while it fails in a more complex case. This finding suggests that directly coordinating actions based on coordination graphs may be more efficient than learning encoded messages. For detailed discussion, please refer to Appendix F.

---

> ### Author Response · Authors · 2021-11-23
> **Response to Reviewer Jx51 (Part II)**
>
>
> **Q2.2: How simple of a task is the "sweet spot" for SOP-CG? As task complexity is altered, when does SOP-CG begin losing its simplicity advantage? When do other models take over? Which measures would you say most greatly determine the task complexity, which in turn, compromises SOP-CG's performance?**
>
>
>
> To illustrate the algorithmic properties of SOP-CG, we propose two measures of task complexity and conduct analysis between SOP-CG and DCG according to these measures. In Appendix G, we design a new didactic example to empirically demonstrate our theoretical implications answering the following questions: when does SOP-CG work better and when does DCG work better?
>
> We will introduce the two measures of task complexity as follows:
>
> - **Problem size in general Dec-POMDPs.**  Solving a general single-agent MDP with state space $S$ and action space $A$ requires the polynomial-time complexity of $|S| \times |A|$ (i.e., its problem size), as shown in Sutton et al. [6]. In MARL tasks, $A = \bigcup A_i$, whose problem size is extended to $|S|\times|A_i|^n$, exponential of the number of agents and the size of local action space, respectively. To analyze the effect of problem size on SOP-CG and CG, we find that the time complexity of exact greedy action selection (solving DCOPs) is exponential-time for DCG (i.e., with a complete graph), and which is polynomial-time complexity for SOP-CG (i.e., with a class of polynomial-time graphs).
>
> - **Structural complexity of coordination within agents.**  This measure is used to represent how much function expressiveness of coordination graph (i.e., which types of graph) is needed to solve a specific task. Note that the function capacity of DCG (line or star), SOP-CG, and DCG (complete graph) forms a hierarchy, i.e., $\mathcal{Q}_{\text{DCG (line or star)}} \subset \mathcal{Q} _ {\text{SOP-CG}} \subset\mathcal{Q} _ { \text{DCG (complete graph)}}$, where $\mathcal{Q}$ indicates the class of value function which can be represented by each algorithm. Thus, we formally design the structural complexity of coordination as the necessary and sufficient function capacity induced by each specific task. Along with his structural complexity increases, larger function capacity of algorithms is required. Moreover, we hypothesize that in the realistic tasks, the coordination structure is sparse [9, 10] and the function class of SOP-CG would be sufficient for an extensive set of situations.
>
>
> Using these two measures of task complexity, we compare SOP-CG with DCG in four cases that either these two complexities are high or low.
>
>
> - **Both two complexities are low.** Both SOP-CG and DCG (complete graph) can easily address this case.
> - **The problem size is large but the structural complexity is relatively low.** which is a common case in real-world tasks [9, 10]. SOP-CG can significantly outperform DCG (complete graph) because of efficient computation of exact greedy action selection without sacrificing its expressiveness in this case. The major strength of SOP-CG is its ability to model the value function using self-organized state-dependent polynomial-time coordination graphs. Solving the DCOPs induced by SOP-CG is computationally efficient but which is expensive for DCG (complete graph) and the approximation of its induced DCOPs. We find that SOP-CG is not sensitive to the problem size, such as the number of agents and individual actions. In comparison, it is NP-hard to find a constant-ratio approximation of the DCOPs induced by complete graphs (i.e., the default implementation of DCG).
> - **The problem size is small but the structural complexity is very high.** In this case, DCG (complete graph) may outperform SOP-CG as it is possible for DCG to compute nearly optimal joint action selection. However, as we hypothesized that this case may not usual in the realistic tasks since multi-agent systems are often large and the sparse coordination structure commonly exists in such systems [9, 10].
> - **Both two complexities are high.** SOP-CG and DCG (complete graph) may perform comparably in this case because they both introduce some approximation in learning. SOP-CG introduces approximation in value function expressiveness while DCG has inaccurate greedy action selection. This case is super hard and would be studied as an important future direction of coordination graph methods.
>
>
> We also design an elaborate didactic example to empirically illustrate the above analysis and its implication. Please refer to Appendix G for details.

---

> ### Author Response · Authors · 2021-11-23
> **Response to Reviewer Jx51 (Part I)**
>
> Thanks for the inspiring comments and additional related work. We provide clarification to your questions and concerns as below. If our response does not fully address your concerns, please post additional questions and we will be happy to have further discussions.
>
> **Q1: The connection to a WWW 2020 paper.**
>
> We thank the reviewer for providing this relevant paper [1]. The connections (i.e., the differences and relevance) between [1] and our paper are summarized as follows:
> - **Different metrics of expressiveness.** Yoon et al. investigates the functionality of graph expressiveness in terms of the order of hyper-edges. By contrast, our paper focuses on the expressiveness of different graph topologies.
> - **Similar motivation.** Both Yoon et al. and our paper aim to study the trade-off between the graph expressiveness and the complexity of solving downstream tasks. In our paper, the downstream task specifically refers to the DCOPs induced by graph-based value factorization.
> - **Different work flow and contributions.** Yoon et al. establish a systematic experiment study to provide an empirical guideline on how to trade off between the graph expressiveness and downstream costs. In comparison, our work proposes an algorithm to break this trade-off for improving MARL performance. Our algorithm is able to improve the function expressiveness without largely increasing the time complexity of downstream DCOPs.
>
> As suggested by the reviewer, in the revised version of our paper, we discuss the relationship of our paper to Yoon et al. [1].
>
> **Q2.1: How did the authors go about designing the didactic examples?**
>
> We conduct the didactic examples in Section 5.1 for empirically analyzing two main contributions of this paper, which (1) introducing the critical role of polynomial-time coordination graphs in the theoretical and empirical perspectives, and (2) using self-organized state-dependent graph to enrich the function expressiveness of static polynomial-time graphs:
> - **The advantages of polynomial-time coordination graphs.** In Figures 3a and 3b, we conduct a comparison between online and offline learning settings. This experiment shows that, by harnessing polynomial-time coordination graphs to obtain accurate greedy action selection, we can both improve the efficiency of exploration and the accuracy of value computation.
> - **The advantages of self-organized coordination graphs.** In Figure 3c, we compare the performance of our method with DCG using static graph topologies. Note that both SOP-CG ($\mathcal{G}_H$/$\mathcal{G}_P$) and DCG (line/star) guarantee accurate greedy action selection. In this case, our method using self-organized graph topologies significantly outperforms the counterpart using static graphs.
>
>
> Furthermore, we conduct another didactic example and discuss the detailed advantages of SOP-CG and DCG in Q2.2.

---

### Author Response · Authors · 2021-11-23
**Summary of the Revision**

Thank you to all the reviewers for your constructive and inspiring comments, which helped us improve our work. Here is a summary of major updates made to the revision:

- We conduct more extensive experiments on 8 StarCraft II micromanagement tasks. The results are shown in Appendix E and F, which strengthen the outperformance of SOP-CG over state-of-the-art coordination graph learning methods.

- We analyze the relative advantages of SOP-CG and DCG and justify our claims by an elaborate didactic example in Appendix G.

- We discuss the worst-case time complexity of SOP-CG in Appendix B and test the real-time training cost in Appendix D.

- We provide detailed clarifications about the literature position, underlying trade-off, and communication mechanism of SOP-CG in Appendix H, I.

We hope that our response and revision address your concerns and questions. We are happy to have further discussions if you have any additional concerns or comments.

---

### Author Response · Authors · 2021-11-30
**To AC: thank you very much for your time and efforts in leading the discussions!**

Dear Area Chair,

We thank all the reviewers for their time and efforts in reviewing our work.

In response to reviewers’ comments, we have provided extensive additional experimental results, illustrations on a new didactic example, time-complexity analysis and comparison, and detailed clarification. Overall, we believe we have addressed all major concerns of reviewers.

As the discussion period is approaching the end, we really appreciate it if you could encourage reviewers to read our response and the revised submission, re-evaluate our work, and post their feedback.

Thank you very much for leading the discussions!

Best,

The Authors

---

### Decision · Program_Chairs · 2022-01-20

**Decision:**

Reject

**Comment:**

Description of paper content:

The paper studies the problem of achieving coordination among a group of agents in a cooperative, multi-agent task. Coordination graphs reduce the computational complexity of this problem by reducing the joint value function to a sum of local value functions depending on only subsets of agents. In particular, the Q-function of the entire system is “expanded” up to second-order in agent interactions: Q = \sum_{i \in [n]} q_i + \sum_{(i,j) \in G} q_{ij}, where the q_i is function of the i-th agent’s history and current action, and q_{ij} is a function of two agents’ histories and current actions. As G does not include higher-order (third and above) terms, the algorithm does not have exponential dependence on the number of agents. If G includes only a subset of pairs of agents, then the computational complexity is reduced to less than quadratic. Since the coordination problem is cooperative, the authors propose a meta-agent (“coordinator”) that selects the graph G in a dynamic (state-by-state) fashion in order to maximize return. The optimization problems of the meta-agent and the sub-agents are performed by deep Q-learning.

Summary of paper discussion:

The critical comment made by one reviewer was: “Going back on that trend now only to pursue the polynomial-time nature of the running algorithm would in my opinion require far more diverse evaluation examples, backed by a stronger motivation highlighting real-world threats of all the other MARL algorithms taking longer than polynomial time. As is, SOP-CG does not contend amazingly against other MARL algorithms that chose the "NP-hard? Curse of dimensionality? Fine. We'll approximate, approximate, approximate." path rather than the "Polynomial time is our topmost priority; function expressiveness can wait." path. That leads me back to the question of why pursue polynomial time at the cost of losing both the function expressiveness and the peak performance….”

Comments from Area Chair:

Looking at the experiments, the number of agents in the empirical problems is not large. For example, there are 15 agents in "Sensor." Any focus on computational complexity at this scale is hard to justify, especially with algorithms that are approximate. It seems favorable at this small scale to use function approximators that can take in all the agents' histories and actions. This obvious baseline is not included in comparisons. It is hard to justify inclusion of this paper in the conference.